# Spatiotemporal Patterns of Aquatic Product Risks in China Based on Entropy-Weighted TOPSIS

**DOI:** 10.3390/foods14244263

**Published:** 2025-12-11

**Authors:** Guangcan Tao, Guoyan Li, Dingfang Pu, Luolin Bao, Su Xu, Hongbo Yang, Kang Hu

**Affiliations:** 1School of Public Health, The Key Laboratory of Environmental Pollution Monitoring and Disease Control, Ministry of Education, Guizhou Medical University, Guiyang 561113, China; tgcan@gyu.edu.cn (G.T.); liguoyan1069@163.com (G.L.); 18886026258@163.com (D.P.); 18385879968@163.com (L.B.); 2School of Food Science and Engineering, Guiyang University, Guiyang 550005, China; xs8515@126.com; 3National Institutes for Food and Drug Control, Beijing 102629, China

**Keywords:** hazardous substances, risk classification, spatial autocorrelation, spatial–temporal distribution characteristics, Pareto principle

## Abstract

This study investigates the risk classification and spatiotemporal evolution patterns of hazardous substances in Chinese aquatic products. The entropy-weighted TOPSIS method was employed to achieve the ranking of hazardous substances and classify their risk levels. A spatial autocorrelation analysis was conducted to explore the spatial distribution patterns of the highest-risk and higher-risk substances in Chinese aquatic products. Risk-adjustment factors were employed to allow us to perform dynamic analyses of the risks in aquatic products across different temporal and spatial contexts. The results indicate that the top three hazardous substances in aquatic products were cadmium, enrofloxacin, and total volatile basic nitrogen; their relative proximity values were 0.707, 0.689, and 0.429, respectively. Cadmium, enrofloxacin, furazolidone metabolites, and chloramphenicol exhibited significant global spatial autocorrelation. The spatiotemporal analysis found that risks in aquatic products were higher during summer and autumn, with the maximum risk value reaching 0.92. The integrated application of the entropy-weighted TOPSIS method, spatial autocorrelation analysis, and risk-adjusted factors provides a novel perspective for risk assessment. The findings support targeted regulation of high-risk substances in Chinese aquatic products and the optimization of seasonal–regional regulatory approaches. It is recommended that regulatory measures and schemes be adjusted in light of the findings, thus providing a scientific foundation for the safety supervision of aquatic products.

## 1. Introduction

Aquatic products encompass animals, plants, and microorganisms sourced from aquatic environments, along with various derivative products obtained through industrial processing [1]. As a vital source of protein, these products contribute nearly 20% of global animal protein consumption and have become a key component of dietary structures [2,3]. They are important sources of high-quality proteins, essential amino acids, fatty acids, and trace elements [4], characterized by high protein content, low fat, easy digestibility, rich mineral content, and appealing taste [5]. These products play a critical role in improving cardiovascular and cerebrovascular health [6], enhancing cognitive function [7], and promoting neural development [8], providing specific nutritional benefits for cardiovascular health in the elderly and intellectual development in children. Asia accounts for over 70% of global aquatic product consumption [9]. As a major producer, consumer, and trader of aquatic products worldwide, China recorded a total aquatic product output of 71.1617 million metric tons in 2023, with imports and exports amounting to 10.5605 million metric tons and a total trade value of USD 44.237 billion, while its per capita annual consumption significantly exceeds the global average [10].

Due to the complexity of the aquatic product supply chain, which encompasses aquaculture, fishing, transportation, processing, and dining consumption, aquatic products are exposed to multidimensional contamination risks. These risks are characterized by the coexistence of traditional and emerging pollutants, as well as intertwined biochemical hazards. The quality and safety of aquatic products have garnered widespread attention, with food safety concerns primarily focusing on heavy metal contamination, drug residues, and biological hazards [11,12]. Heavy metals are toxic and potentially carcinogenic. They accumulate in aquatic systems, leading to bio-accumulation even at low exposure concentrations, and can damage multiple organs, including the nervous system, liver, lungs, kidneys, stomach, skin, and reproductive system [13]. For instance, arsenic can cause reproductive disorders and damage to the nervous system; lead may impair the nervous system, disrupt brain activity, and impair bone marrow function; cadmium is linked to adverse effects on bones, kidneys, and reproductive systems [14,15]. As aquaculture expands in scale, stocking densities have increased, which in turn leads to frequent outbreaks of various diseases. This has also been coupled with the increased overuse of drugs, non-compliant medication use, and the use of banned substances, resulting in the residues of these substances in aquatic products and widespread concern [16,17]. Common drug residues in aquatic products primarily include nitrofuran metabolites, chloramphenicol, quinolones, sulfonamides, and tetracyclines [18,19,20]. These residues pose significant health risks to humans, including the development of antibiotic resistance, carcinogenic effects, teratogenic effects, allergic reactions, and disruption of normal gut microbiota [21,22]. In response to the aforementioned risks, scholars have progressively developed a multidimensional assessment framework. Chen et al. [23] comprehensively evaluated heavy metal contamination in aquatic products and the associated health risks by integrating single-factor pollution indexes, composite pollution indexes, and safety assessments. Ku et al. [24] proposed a decision tree model for heavy metal hazards to assess the potential health risks of 12 heavy metals in seven categories of aquatic products. This study integrates statistical data from the fisheries in Taiwan and national food consumption data, analyzing heavy metal concentrations in 556 cooking aquatic product samples; additionally, it conducts risk assessments using estimated daily intake (EDI) and hazard quotients (HQs). Liu et al. [25] determines indicator weights using the Analytic Hierarchy Process (AHP), and calculates risk indices using comprehensive index and multiplicative synthesis. By taking data from Guangzhou as an example, the annual risk indices and grades are calculated to assess food safety conditions and validate the index methods. The final risk assessment indicator system comprises five primary indicators, thirteen S-level indicators, and thirteen chemical contaminants, covering hazards, vulnerability, monitoring, consumption patterns, public sentiment, and regulatory oversight. Existing studies have laid a foundation for assessment methodologies; however, most studies rely on cross-sectional data, lacking any analysis of risk dynamics over time and neglecting spatiotemporal variations, thereby failing to reveal patterns of risk evolution. Methods remain subjective, with approaches such as the AHP relying on expert weighting and thereby compromising assessment objectivity. Analyses are incomplete, lacking multi-pollutant, cross-regional risk transmission assessments. This constrains the precision of aquatic product risk warnings and the effectiveness of regulatory strategies, failing to meet the requirements for a “data-driven” regulatory transformation. A dynamic assessment framework is urgently needed.

Based on the aforementioned issues, the present study proposed a three-dimensional assessment framework integrating time, space, and hazardous substances. Based on the supervision data of 1.04 million batches of aquatic products from China’s 31 provinces between 2021 and 2023, four categories of indicators were consolidated: non-compliance rate, detection rate, qualification degree, and hazard degree. Methodologically, the entropy weighting method was employed to objectively assign weights, overcoming the limitations of subjective preferences inherent in traditional AHP. By combining the TOPSIS model, the framework quantified the priority of hazardous substances and achieved dynamic ranking by calculating the proximity of each sample to the ideal solution. Based on the Pareto principle, risk substances were categorized into distinct risk levels (lowest, lower, medium, higher, and highest). Spatiotemporal distribution patterns were analyzed to assess monthly trends in non-compliance rates for risk substances between 2021 and 2023, verifying seasonal risk patterns. Global and local spatial autocorrelation analyses were employed to explore geographical distribution characteristics of high-risk substances and identify high-risk clusters. A spatiotemporal risk dynamics model was constructed, incorporating provincial and monthly risk adjustment factors to calculate the final spatiotemporal risk values for aquatic products. High-risk “time–area” heatmaps were generated to investigate risk trends across time and space. This model aimed to reveal the spatiotemporal distribution patterns of hazardous substances in aquatic products and the risk transmission pathways within the supply chain. It provides algorithmic support for constructing a comprehensive, intelligent, and dynamic monitoring network spanning the entire chain “from farm to fork”, offering scientific basis for the safety supervision of aquatic products. It drives the transformation of aquatic product safety governance from an experience-driven approach to a data-driven paradigm, facilitating the transition from extensive regulation to precision-based management.

## 2. Materials and Methods

### 2.1. Data Sources and Study Methods

The data in this study were sourced from the publicly released supervision and inspection data of aquatic products by the market supervision administrations in various provinces, municipalities, and autonomous regions (excluding Taiwan, Hong Kong, and Macao) between 2021–2023. This data strictly adheres to the supervision and sampling principles of the national market regulatory system: the sampling design covers all stages of distribution and catering, employing stratified random sampling to cover all 31 provinces in China and various types of establishments. Both samples and testing items are determined in accordance with national standards, ensuring the data’s representativeness, authenticity, and standardization nationwide. Based on the research objectives, the supervision and inspection data of aquatic products were systematically organized, resulting in the retention of nine key items: sampling location, sampled province, sampling date, sampling stage, the regional type of sampled entity, inspection items, sampling results, and the determination of sampling results. Four types of indicators were selected for analysis: non-compliance rate, detection rate, qualification degree, and hazard degree [26]. The rankings of risk substances were determined using the entropy-weighted TOPSIS method, and risk classification was conducted in accordance with the Pareto principle.

### 2.2. Indicator Selection

Based on information contained within aquatic product supervision sampling data, risk indicators comprising non-compliance rates, detection rates, qualification degree, and hazard degree—combined by food category and testing parameters—were established to evaluate aquatic food safety risks [27]. Detection rates and non-compliance rates are the indicators most commonly employed for risk assessment. However, relying solely on these two metrics for aquatic product risk evaluation inherently reduces data utilization efficiency. Moreover, due to the vast volume of supervisory sampling data, instances of non-compliance are far fewer than compliant data. Consequently, further analysis of compliant data with detectable values is required. Hence, the qualification degree indicator was introduced. As different food categories correspond to distinct types of testing items, and different testing item types relate to varying risk substances, each risk substance carries a different severity level. Therefore, the hazard degree indicator was introduced to assign values to different risk substances, with higher values indicating greater corresponding risk.
(1)Non-compliance rate: When test values are greater than or equal to national or international maximum permitted levels, the sample is deemed non-compliant. This refers to the percentage of non-compliant results detected relative to the total number of tests conducted.(2)Detection rate: This represents the percentage of samples yielding detectable levels of the risk substance relative to the total number tested.(3)Qualification degree: For risk indicators deemed compliant in testing results, the closer the detected value approaches the national standard limit, the greater the likelihood of non-compliance and the higher the risk. Conversely, the closer the detected value approaches the laboratory’s detection limit—that is, the further it is from the national standard limit—the lower the likelihood of non-compliance and the lower the risk.(4)Hazard degree: This denotes the gravity of a hazard factor’s impact on consumer health, typically quantified using three indicators—health guidance values, carcinogenicity, and median lethal dose (LD_50_) [28]. Where all three risk indicators are assigned values for a given hazard, the highest value is selected as the severity score. The severity scoring table for food hazards is provided in the Appendix A.

### 2.3. Analytical Methods

#### 2.3.1. Construction of Risk Classification Model Based on the Entropy-Weighted TOPSIS Method

The entropy-weighted TOPSIS model is an integrated approach that combines the entropy weight method with the TOPSIS technique. The entropy weight method is employed to determine the weights of various indicators by analyzing the information they provide. The TOPSIS method can be used to calculate the distances between each indicator and its ideal solution, thereby assessing the proximity of each evaluation object to the optimal level. The TOPSIS method has demonstrated strong applicability in comprehensive evaluations across multiple food safety domains in recent years. Often combined with other weight-determination methods such as AHP and entropy weighting, it is employed for risk grading assessments by food-safety-related manufacturers [29], for risk management in food supply chains [30], for the optimization of food formulations [31], for the ranking of international markets based on the rigor of food safety measures [32], and so on. This approach assists decision-makers in conducting scientific and objective comprehensive evaluations. The combination of the entropy weighting method and the TOPSIS method avoids the subjectivity associated with manual weight assignment, reducing the potential for bias from human judgment. Consequently, it yielded evaluation results that were more aligned with objective reality. The model boasted several advantages such as strong interpretability, intuitive geometric interpretation, and flexible computation [33,34].

##### Calculation of Weights Using the Entropy Weight Method

For n samples and k indicators, X_ij_ represents the jth indicator for the ith sample (where i = 1, 2, …, n; j = 1, 2, …, k),
$D_{\mathrm{ij}}$ denotes the standardized value of each indicator. The matrix X was obtained from i risk factors and j indicators, as shown in Equation (1).
(1)$$\left.X=\left[\begin{array}{cccc} x_{11} & x_{12} & \ldots & x_{1j}\\ x_{21} & x_{22} & \ldots & x_{2j}\\ \ldots & \ldots & \ldots & \ldots \\ x_{i1} & x_{i2} & \ldots & x_{\mathrm{ij}} \end{array}\right.\right]$$

We conducted dimensionless processing on the original matrix to eliminate the interference of differing units of measurement, yielding the matrix
$D_{\mathrm{ij}}$ (Equation (2)). Max (X_j_) is the maximum value for each indicator, while min (X_j_) is the minimum value. Should zero values appear in the standardized matrix after processing, to ensure meaningful data handling, all dimensionless data are shifted by a minimum unit value.
(2)$$D_{ij}=\left\{\begin{array}{c} \frac{x_{ij}-min(x_{j)}}{max(x_{j})-min(x_{j})}\;\mathrm{Positive}\;\mathrm{index}\\ \frac{max(x_{j})-x_{j}}{max(x_{j})-min(x_{j})}\;\mathrm{Negative}\;\mathrm{index} \end{array}\right.$$

Then, we calculated the feature weight or contribution of the i-th item under the j-th indicator (Equation (3)):
(3)$$P_{\mathrm{ij}}\;=\;\frac{D_{\mathrm{ij}}}{\sum_{j=1}^{n}D_{\mathrm{ij}}}$$

The information entropy was calculated as follows (Equations (4) and (5)):
(4)$$e_{j}\;=\;-k\sum_{j=1}^{n}P_{\mathrm{ij}}\ln P_{\mathrm{ij}}$$
(5)$$K=\frac{1}{\ln(n)}$$

The weights were calculated as follows (Equations (6) and (7)):
(6)$$g_{j}\;=\;1\;-\;e_{j}$$
(7)$$W_{j}=\frac{g_{j}}{\sum g_{j}}$$

##### Calculation of Relative Proximity Using the TOPSIS Method

Using the weighted standardized matrix, we multiplied matrix D_ij_ by the obtained indicator weights W_j_ to obtain matrix U_ij_, as shown in Equation (8):
(8)$$U_{\mathrm{ij}}\;=\;D_{\mathrm{ij}}\;\times \;W_{j}$$

The optimal value vector Ui+ and the suboptimal value vector Ui- for each indicator were determined as follows (Equations (9) and (10)):
(9)$$U_{i}^{+}\;=\;\max\left\{U_{1j}\;,\;U_{2j}\;,\;\ldots ,\;U_{\mathrm{nj}}\;\right\}$$
(10)$$U_{i}^{-}=\min\left\{U_{1j}\;,\;U_{2j}\;,\;\ldots ,\;U_{\mathrm{nj}}\;\right\}$$

The distances of each evaluation object from the optimal value vector and the suboptimal value vector were calculated as follows (Equations (11) and (12)):
(11)$$D_{i}^{+}=\sqrt{\sum_{j=1}^{n}W_{j}\;\times \;{\left(U_{\mathrm{ij}}-U_{j}^{+}\right)}^{2}}$$
(12)$$D_{i}^{-}=\sqrt{\sum_{j=1}^{n}W_{j}\;\times \;{\left(U_{\mathrm{ij}}-U_{j}^{-}\right)}^{2}}$$

The relative proximity was calculated, with values ranging from [0, 1], where a larger value indicated greater risk associated with the hazard, as shown in Equation (13):
(13)$$C_{i}\;=\;\frac{D_{i}^{-}}{D_{i}^{+}\;+\;D_{i}^{--}}$$

#### 2.3.2. Pareto Principle

The core theoretical basis of the Pareto principle is that 80% of outcomes are typically determined by 20% of key factors. This theory has been widely applied in fields such as risk classification and resource optimization. We used the Pareto principle to classify relative proximity values into five tiers: [0, 10%) was designated as lowest risk, [10%, 40%) was designated as lower risk, [40%, 70%) was designated as medium risk, [70%, 90%) was designated as higher risk, and [90%, 100%] was designated as highest risk [27].

#### 2.3.3. Spatial Autocorrelation Analysis

##### Global Spatial Autocorrelation Analysis

Spatial analysis was employed to identify clustered areas and examine geographic spatial variations. By describing the spatial distribution characteristics of non-compliance rates for high-risk and higher-risk substances in aquatic products, we utilized Moran’s I statistic, which ranged between −1 and 1. Values close to 1 indicated positive spatial autocorrelation, values close to −1 indicated negative spatial autocorrelation, and 0 indicated random distribution. Moran’s I was calculated and a hotspot analysis was performed to determine the locations of the clusters [35]. Typically, a weight (W_ij_) is associated with each pair (X_i_, X_j_) to quantify spatial patterns, as shown in Equation (14):
(14)$$I\;=\;\frac{n\sum_{i=1}^{n}\sum_{j=1}^{n}W_{\mathrm{ij}}\left(x_{i}-\overset{‾}{x}\right)\left(x_{j}-\overset{‾}{x}\right)}{\sum_{i=1}^{n}\sum_{j=1}^{n}W_{\mathrm{ij}}\left(x_{i}-\overset{‾}{x}\right)}$$

In the formula, n represents the number of provinces, municipalities, and autonomous regions in China; X_i_ and X_j_ denote the non-compliance rates of a certain risk factor in regions i and j across the country, respectively;
$\overset{‾}{x}$ represents the average non-compliance rates; W_ij_ is the spatial weighting matrix.

##### Local Spatial Autocorrelation Analysis

Local Moran’s I was employed to determine local spatial autocorrelation, identifying spatial clusters with similar neighboring characteristics and outliers, as shown in Equation (15):
(15)$$I\;=\;\frac{\left(x_{i}-\overset{‾}{x}\right)}{S^{2}}\sum_{j=1}^{n}W_{\mathrm{ij}}\left(x_{j}-\overset{‾}{x}\right)$$

In the formula, S^2^ denotes the variance of X_i_ or X_j_, while the other components are interpreted similarly to global Moran’s I.

#### 2.3.4. Spatiotemporal Analysis

Through statistical analysis, the non-compliance rates of aquatic products across provinces and each month were yielded, and thus the risk adjustment factors for different provinces and months were obtained, as shown in Equations (16)–(18) [36]. In the formula, P is the risk adjustment factor for a province; P_f_ is the number of non-compliant cases in that province; Q_f_ represents the total number of non-compliant cases for food products; M denotes the risk adjustment factor for a month; M_f_ signifies the total number of non-compliant cases across all provinces for that month. The maximum relative proximity was taken as the baseline risk for the food, to which a risk adjustment factor was added to calculate the final spatiotemporal risk value for the aquatic product.
(16)$$P=1+\frac{P_{f}}{Q_{f}}$$
(17)$$M=1+\frac{M_{f}}{Q_{f}}$$
(18)$$R={C_{i}}_{\max}\times \;P\;\times \;M$$

#### 2.3.5. Statistical Analysis

In this study, the inspection data of aquatic products from 2021 to 2023 were organized using Microsoft Excel 2019 (Microsoft Corporation, Redmond, WA, USA). Origin 2021 (OriginLab Corporation, Northampton, MA, USA) and ArcGIS 10.8 (Environmental Systems Research Institute, Redlands, CA, USA) were used to create maps. We used a significance level of α = 0.05. The Z-test was employed for spatial autocorrelation analysis. The statistical assumptions made included a spatial autocorrelation analysis and the Pareto principle classification. Since the data in this study are ordered categorically or skewed, nonparametric statistical methods are employed.

## 3. Results

### 3.1. Detection Results of Risk Substances in Aquatic Products

After processing the inspection data of aquatic products from 2021 to 2023, a total of 1,044,312 valid samples were obtained. The inspection items encompassed six categories, comprising 34 items in total: heavy metals, additives, veterinary drugs, prohibited drugs, quality indicators, and organic pollutants. These inspections covered both distribution and catering stages. The non-compliance rate, detection rate, qualification degree, and hazard degree for all inspection items across different stages were assessed. Utilizing the entropy weight method, the weights for the four indicators—non-compliance rate, detection rate, qualification degree, and hazard degree—were calculated for both the distribution and catering stages. The weight coefficients (w) for hazard degree, qualification degree, detection rate, and non-compliance rate in the distribution stage were 5.46%, 17.18%, 35.45%, and 41.91%, respectively, while those in the catering stage were 5.72%, 8.14%, 36.89%, and 49.25%, respectively. Subsequently, the TOPSIS model was employed to comprehensively calculate the proximity of each sample to the ideal solution, resulting in the dynamic ranking presented in Table 1. Among the risk substances identified in both the distribution and catering stages, cadmium, enrofloxacin, and total volatile basic nitrogen ranked highest in proximity.

Relative proximity was classified according to the Pareto principle, with the classification range detailed in Table 2 and specific classification results presented in Table 3. During the distribution stage, cadmium, enrofloxacin, and total volatile basic nitrogen were identified as the highest-risk substances. Sulfur dioxide, diazepam, methylmercury, malachite green, furazolidone metabolites, chloramphenicol, nitrofurazone metabolites, sodium pentachlorophenate, metronidazole, nitrofurantoin metabolites, furaltadone metabolites, sarafloxacin, and ofloxacin were categorized as higher-risk substances. Polychlorinated biphenyls, danofloxacin, flumequine, difloxacin, oxolinic acid, inorganic arsenic, chromium, and histamine fell into the medium-risk category. Lead, deltamethrin, cypermethrin, trimethoprim, florfenicol, and sulfonamides were classified as lower-risk substances, while pefloxacin, oxytetracycline/chlortetracycline/tetracycline (sum), norfloxacin, and lomefloxacin were assigned to the lowest-risk category. In the catering stage, cadmium, enrofloxacin, and total volatile basic nitrogen were classified in the highest-risk category, while methylmercury and sodium pentachlorophenate posed an elevated risk. Inorganic arsenic, chromium, nitrofurazone metabolite, furazolidone metabolite, diazepam, malachite green, chloramphenicol, metronidazole, lead, furaltadone metabolites, nitrofurantoin metabolites, and sarafloxacin were categorized as medium-risk substances. Compounds such as ofloxacin, oxolinic acid, difloxacin, pefloxacin, norfloxacin, danofloxacin, flumequine, lomefloxacin, deltamethrin, polychlorinated biphenyls, cypermethrin, florfenicol, oxytetracycline/chlortetracycline/tetracycline (sum), sulfur dioxide, and histamine were considered lower-risk. Additionally, trimethoprim and sulfonamides (total amount) were identified as the lowest-risk substances. Overall, the risk of various risk substances in aquatic products was higher in the distribution stage than that in the catering stage.

An analysis of non-compliant aquatic products (Figure 1) revealed that the highest non-compliance rate was observed in seawater crabs (3.19%), followed by other aquatic products (2.25%), and the lowest rate was freshwater crabs (0.04%). A detailed examination of non-compliant items across various categories of aquatic products was conducted, with the proportions of each non-compliant item by category illustrated in Figure 2. The primary non-compliant items in freshwater crab were identified as heavy metals and prohibited drugs; in freshwater shrimp, shellfish, and other aquatic products, the main items were heavy metals, veterinary drugs, and prohibited drugs. For seawater fish, the predominant issues were heavy metals, veterinary drugs, prohibited drugs, and quality indicators, while seawater shrimp primarily exhibited concerns related to heavy metals, veterinary drugs, prohibited drugs, and additives. Seawater crabs were mainly associated with heavy metals, and in freshwater fish, veterinary drugs and prohibited drugs were the primary concerns. The distribution of non-compliant items detected in different categories of aquatic products is presented in Table 4. For seawater crabs, cadmium was the primary concern, while other aquatic products predominantly showed issues with enrofloxacin, furazolidone metabolites, and nitrofurazone metabolites; for seawater shrimp, it was mainly cadmium and enrofloxacin; freshwater fish was mostly associated with enrofloxacin, diazepam, and malachite green, while shellfish primarily contained chloramphenicol and cadmium; both freshwater shrimp and seawater fish had significant levels of enrofloxacin and furazolidone metabolites, whereas freshwater crabs primarily contained cadmium, nitrofurazone metabolites, and furazolidone metabolites.

An analysis of the non-compliant items in aquatic products revealed that veterinary drugs constituted the largest proportion at 53.74%, followed by heavy metals at 25.38%. No non-compliant cases were detected for organic pollutants. Based on the risk classification of aquatic products and the identified non-compliant items in aquatic products (Figure 3), the primary focus should be on veterinary drugs and heavy metals. Among veterinary drugs (Figure 4a), particular attention should be paid to the illegal use of enrofloxacin, which accounted for 92.96% of the non-compliant cases. Regarding heavy metals, only cadmium was found to be non-compliant. In combination with the risk classification, significant attention should be paid to cadmium and methylmercury. The inspection items under prohibited drugs (Figure 4b) all presented relatively high risks and require close attention. Non-compliance in quality indicators was detected for histamine and total volatile basic nitrogen; considering the risk classification, total volatile basic nitrogen requires particular attention. In the category of additives, only sulfur dioxide was found to be non-compliant, which should be closely monitored in light of the risk classification.

### 3.2. Temporal Distribution of Risky Substances in Aquatic Products

A temporal analysis of non-compliant cases in aquatic products from 2021 to 2023 (Figure 5) reveals a general trend of gradual increase in non-compliance across different months, exhibiting a distinct seasonal pattern. The non-compliance rate was notably higher in summer and autumn compared to winter and spring. Overall, the non-compliance rate in the second half of the year was higher than that in the first half, exhibiting a gradual upward trend, though the overall fluctuation remained relatively small. In 2021, the lowest non-compliance rate was 0.12% in February, and the highest was 1.73% in December, indicating significant fluctuation. In 2022, the lowest rate was 0.44% in December, and the highest was 1.05% in October. In 2023, the non-compliance rate was lowest in December at 0.23% and highest in November at 0.58%, demonstrating relatively minor fluctuation.

#### 3.2.1. Veterinary Drugs

A temporal analysis was conducted on the non-compliance rate of veterinary drugs in aquatic products (Figure 6a). The primary non-compliant substances comprised five categories: enrofloxacin, sulfonamides, trimethoprim, florfenicol, and ofloxacin. Notably, enrofloxacin exhibited non-compliance throughout the entire year, peaking in December with a rate of 4.83%. Trimethoprim also reached its highest non-compliance rate of 2.50% in December. Additionally, florfenicol recorded a non-compliance rate of 1.03% in May, which warrants particular attention. Other veterinary drugs showed sporadic instances of non-compliance, maintaining relatively stable rates overall.

#### 3.2.2. Heavy Metals

A temporal analysis of non-compliant cases related to heavy metals in aquatic products (Figure 6b) indicated that cadmium was the only heavy metal that was non-compliant. The non-compliance rate fluctuated significantly throughout the year, showing an upward trend. It peaked at 5.63% in December and was lowest at 0.92% in February. Continuous attention to cadmium is required.

#### 3.2.3. Prohibited Drugs

A temporal analysis was conducted on the non-compliance rate of prohibited drugs in aquatic products (Figure 6c). The primary non-compliant substances identified were furazolidone metabolites, malachite green, chloramphenicol, nitrofurazone metabolites, sodium pentachlorophenate, and diazepam. Notably, diazepam exhibited the highest non-compliance rate in November, reaching 1.14%; chloramphenicol peaked in May at 0.72%; and malachite green reached its highest level of 0.71% in April. Other prohibited drugs displayed sporadic non-compliance with relatively stable rates. Overall, the non-compliance of prohibited drugs was minimally affected by seasonal variations.

#### 3.2.4. Quality Indicators

A temporal analysis was conducted on the non-compliance of quality indicators in aquatic products (Figure 6d), and the only non-compliant item among the quality indicators was total volatile basic nitrogen. Overall, the risk of non-compliance for total volatile basic nitrogen was relatively low. In May, the highest non-compliance rate was 0.43%, which may be attributed to temperature elevation causing excessively high values in quality indicators, requiring targeted monitoring.

#### 3.2.5. Additives

A temporal analysis of non-compliance related to additives in aquatic products (Figure 6e) showed that the only non-compliant additive was sulfur dioxide. The non-compliance rate fluctuated significantly, reaching a peak of 3.78% in May. The non-compliance rate remained relatively high from September to November, at 1.44%, 1.89%, and 1.51%, respectively.

### 3.3. Spatial Distribution of Risk Substances in Aquatic Products

An analysis was conducted on the non-compliant sample data of aquatic products across provinces during 2021–2023. A classification map of the non-compliance rate was generated using the equal interval method in ArcGIS (Figure 7), illustrating the overall national non-compliance rate of aquatic products (a), and non-compliance rate of heavy metals (b), quality indicators (c), additives (d), veterinary drugs (e), and prohibited drugs (f). The analysis of the overall non-compliance rate across cities revealed a discernible pattern, with higher rates observed in the eastern regions and lower rates in the western areas. Additionally, coastal regions exhibited higher non-compliance rates compared to inland areas. High-risk zones were predominantly located in coastal cities of East and South China, such as the Yangtze River Delta and Pearl River Delta. Meanwhile, provinces in the central and western regions, such as Qinghai and Tibet, were classified as low-risk areas. The non-compliance rates in Zhejiang, Chongqing, and Jiangxi were higher than those in other provinces, at 1.66%, 1.52%, and 1.48%, respectively, with Zhejiang being the province with the highest number of non-compliant aquatic products nationwide. An analysis of non-compliance rate of heavy metals across cities indicated that high-risk areas were concentrated in northern cities, medium risk in coastal cities, and low risk in the northwest region. Regarding the non-compliance rate of quality indicators across cities, there was a scattered distribution without significant large-scale clustering. High-risk areas were sporadically distributed in the northeastern industrial zone, the North China Plain, and the Sichuan Basin, while most regions across the country exhibited low risk. For the non-compliance rate of additives across cities, it revealed isolated high-value points without marked large-scale clustering and no evident spatial patterns.

By analysis of the non-compliance rate of veterinary drugs across cities, it was found that non-compliance mainly occurred in aquaculture-intensive areas in South and East China (such as Guangdong, Fujian, and Jiangsu), while the risk was relatively low in northern pastoral areas and southwestern mountainous regions. An analysis of the non-compliance rate of prohibited drugs across cities displayed that high-risk areas were identified in provinces (autonomous regions and municipalities) such as Beijing, Sichuan, Guangxi, and Zhejiang, while most cities across the country exhibited relatively low risks.

#### 3.3.1. Spatial Autocorrelation Analysis

According to Table 3 that classifies the risk substances in aquatic products, the hazards with the highest and higher risks were selected as the focus (the non-compliance rates of various risk substances across different provinces are provided in the Appendix A). Among them, there were no non-compliant samples for methylmercury or sarafloxacin; furaltadone metabolites was found to have non-compliance issues only in Jiangxi; nitrofurantoin metabolites showed non-compliance only in Inner Mongolia and Jiangxi. Therefore, we analyzed the spatial distribution patterns of the following risk substances: cadmium, enrofloxacin, total volatile basic nitrogen, sulfur dioxide, diazepam, malachite green, furazolidone metabolites, chloramphenicol, nitrofurazone metabolites, sodium pentachlorophenate, metronidazole, and ofloxacin.

##### Global Spatial Autocorrelation Analysis

A spatial autocorrelation analysis was conducted on the non-compliance rate of 12 risk substances in aquatic products across provinces in China. The global Moran’s I value for each risk substance in aquatic products were obtained (Table 5). Among these, the *p*-values for total volatile basic nitrogen, sulfur dioxide, diazepam, malachite green, nitrofurazone metabolites, sodium pentachlorophenate, metronidazole, and ofloxacin were all greater than 0.05, indicating a lack of global spatial autocorrelation in the non-compliance rate of these eight risk substances in aquatic products across provinces. However, local spatial correlation may still exist. The *p*-values for cadmium, enrofloxacin, furazolidone metabolites, and chloramphenicol in aquatic products across provinces were less than 0.05, suggesting that these four risk substances exhibited global spatial correlation in non-compliance rate across provinces, reflecting a clustered distribution pattern.

##### Local Spatial Autocorrelation Analysis

When there is a global Moran’s I index with statistical significance (*p* < 0.05), further analysis of the local Moran’s I index can be conducted to investigate whether there is regional spatial clustering in the study subjects. A local spatial autocorrelation analysis was performed on cadmium, enrofloxacin, furazolidone metabolites, and chloramphenicol in aquatic products across provinces in China. First of all, the *p*-values of the local Moran’s I index were analyzed. If the *p*-value is less than 0.05, it indicates the presence of a regional spatial correlation; conversely, if the *p*-value is greater than 0.05, it indicates the absence of spatial correlation (Table 6). A high–high clustering signifies that both the province and its neighboring provinces were at a “high level”; a low–high clustering meant that the province was at a “low level”, but its neighboring provinces were at a “high level”; a low–low clustering indicated that both the province and its neighboring provinces were at a “low level”; a high–low clustering denoted that the province was at a “high level”, but its neighboring provinces were at a “low level”. For the Moran scatter plot, the *X*-axis (z-score of outliers) represented the distance between data values and the mean; therefore, the data points located further to the right indicated a higher non-compliance rate. The *Y*-axis (Spatial Lag) represented the spatial lag value, with higher values indicating that the neighbors (surrounding areas) of the study subject had higher non-compliance rate.

From Figure 8, the compliance rate for cadmium exhibited significant low–low clustering in Chongqing, Sichuan, Guizhou, Yunnan, Tibet, Shaanxi, Gansu, Qinghai, and Xinjiang. In contrast, it showed significant high–high clustering in Beijing, Tianjin, Hebei, Inner Mongolia, Liaoning, Jilin, Heilongjiang, and Shandong. Obvious low–high clustering was observed in Shanxi. For enrofloxacin, the compliance rate displayed significant low–low clustering in Hebei, Shanxi, Inner Mongolia, and Liaoning, and significant high–high clustering in Shanghai, Jiangsu, Zhejiang, Anhui, Fujian, Jiangxi, Hubei, and Hunan. Additionally, marked low–high clustering was observed in Guizhou. Regarding the non-compliance rate of furazolidone metabolites, there was remarkable low–low clustering in Hebei, Inner Mongolia, and Jilin, and significant high–high clustering in Fujian, Shanghai, Guangdong, Hainan, and Zhejiang, while remarkable low–high clustering was observed in Jiangxi and Jiangsu. For chloramphenicol, significant low–high clustering was detected in Yunnan and Guizhou, significant high–low clustering in Sichuan, and significant high–high clustering in Guangdong and Hainan.

#### 3.3.2. Sampling Location Analysis

The sampling locations for aquatic products included urban areas, township areas, tourist attractions (urban), tourist attractions (townships), vicinity of schools (townships), and vicinity of schools (urban). Non-compliant aquatic product samples were primarily found in urban areas (Figure 9), accounting for 81.34%, while township areas accounted for 16.04%.

#### 3.3.3. Sampling Venue Analysis

The inspection of aquatic products included two sampling stages: distribution and catering. The highest incidence of non-compliance was observed in the distribution stage, accounting for 85.71%, followed by the catering stage at 14.29% (Figure 10). An analysis of the sampling venues (Figure 11) revealed that, within the distribution stage, non-compliant cases predominantly occurred in farmers’ markets and supermarkets, representing 38.75% and 26.48%, respectively. Fewer non-compliant cases were identified in wholesale markets, other locations, vegetable markets, online purchases, small groceries, and shopping malls, accounting for 10.39%, 11.16%, 5.55%, 5.41%, 1.99%, and 0.27%, respectively. Therefore, it is crucial to pay special attention to the quality and safety of aquatic products in farmers’ markets and supermarkets. In the catering stage, nearly half of the non-compliant cases were found in medium-sized restaurants, comprising 45.69%. The remaining non-compliant cases were distributed across small restaurants, large restaurants, other locations, extra-large restaurants, snack bars, and catering service providers, accounting for 33.68%, 15.81%, 1.95%, 1.33%, 0.51%, and 1.03%, respectively. Special attention should be given to the quality safety of aquatic products in small- and medium-sized restaurants.

### 3.4. Spatiotemporal Analysis of Risk Substances in Aquatic Products

According to the evaluation results obtained through the entropy-weighted TOPSIS method, the maximum relative proximity value
$C_{i}$ of cadmium was found to be 0.707, which serves as the baseline risk for aquatic products. After incorporating the risk adjustment factor, the final spatiotemporal risk values for aquatic products were calculated, as illustrated in Table 7. Based on the monthly final spatiotemporal risk values for each province, a heatmap (Figure 12) was generated to illustrate the risk variations of aquatic products across different provinces and time periods by changes in color intensity, with red indicating high risk and blue indicating low risk. The figure indicates that the range of risk values was concentrated between 0.71 (low risk) and 0.92 (high risk), with a relatively narrow span, indicating that, while there were regional differences in the risk of aquatic products nationwide, the overall risk was controllable. From August to November, the risks were higher across provinces (municipalities, autonomous regions), while from December to April, the overall risk was lower. Notably, Chongqing, Jiangxi, and Zhejiang exhibited higher risks throughout the year, whereas Tibet, Ningxia, and Gansu demonstrated lower annual risks.

## 4. Discussion

### 4.1. Core Risk Substances and Their Health Risks

The findings of this study identified the presence of high-risk substances, including cadmium, enrofloxacin, and total volatile basic nitrogen, in aquatic products utilizing the entropy-weighted TOPSIS method. The substances associated with higher risks include sulfur dioxide, diazepam, methylmercury, malachite green, nitrofuran compounds, chloramphenicol, sodium pentachlorophenate, metronidazole, sarafloxacin, and ofloxacin. An analysis of monitoring data for heavy metals in Chinese aquatic products has found that, among the different aquatic product–heavy metals combinations detected, the sea crab–cadmium combination posed the highest risk [37], which aligns with the findings of the present study’s findings that seawater crabs exhibited the highest proportion of non-compliance rates, with cadmium posing the highest risk. Ding et al. [38] conducted testing on 300 batches of commercially available aquatic product samples for veterinary drug residues and heavy metals, and revealed that key risk parameters included cadmium, malachite green, enrofloxacin, nitrofurazone metabolite, and ofloxacin, which are consistent with the results of the present study. This confirms the reliability of the aquatic product classification model developed herein. Based on the risk classification, it is essential to enhance inspections for cadmium, enrofloxacin, total volatile basic nitrogen, sulfur dioxide, diazepam, and methylmercury in the distribution stage of aquatic products, and to increase inspections for cadmium, enrofloxacin, volatile basic nitrogen, methylmercury, and sodium pentachlorophenate in the catering stage.

Among heavy metals, cadmium warrants particular attention. Cadmium in the environment exhibits a bio-amplification effect, which can accumulate through food sources and subsequently transfer to humans via the food chain. Fish, crustaceans, and bivalves are the most common aquatic products, and are also among the main sources of cadmium exposure [39,40]. These aquatic products can accumulate cadmium in their tissues through body surface permeation and ingestion, and absorb cadmium from water [41]. A study from Laizhou Bay, China, indicates that cadmium is the second most abundant metal and the most easily accumulated element in seafood, with crabs exhibiting a much higher cadmium enrichment capacity than other aquatic products [42]. Cadmium contamination exhibits distinct species specificity and regional variation, reflecting the persistent impact of industrial emissions on coastal ecosystems [43]. Bioavailability refers to the extent to which nutrients, toxins, or other substances are available for utilization or accumulation by the body following exposure. Studies indicate that cadmium bioavailability varies across species and processing methods. Cadmium bioavailability varies significantly across different foods. Except for squid and scallops, bioavailability values are generally <1, potentially related to cadmium forms in internal organs. Cooking processes affect food bioavailability, while frying further increases absorption. Traditional risk assessment models may underestimate actual exposure levels [44]. Studies in the Niger Delta confirm that, while cadmium levels in local clams and tilapia partially fall below regulatory limits, long-term consumption still elevates the integrated hazard index. This highlights the hidden nature of geographical variations and cumulative risks [45]. It demonstrates that, even when pollutant concentrations remain below thresholds, regional dietary habits and food chain enrichment effects can still lead to cumulative health risks. Heavy metals can move upward along the food chain, posing potential health risks to humans [1]. Given that food serves as a significant conduit for the transfer of heavy metals from the environment to the human body, it is imperative to regulate the levels of heavy metals in aquatic products to mitigate potential health risks. Long-term consumption of high-risk species should be avoided, and it is advisable to choose freshwater fish or small-sized shellfish with low bio-accumulation capacity for heavy metals. Soil and water quality investigations for heavy metal pollution should be conducted in aquaculture areas to trace and remediate agricultural land contamination. Strict limitations should be imposed on the discharge of industrial wastewater and agricultural non-point-source pollution to reduce heavy metals in water bodies, thereby lowering the bio-accumulation risk of aquatic products from the source [46].

Among veterinary drugs, enrofloxacin should be a primary focus, while particular attention should be given to high-risk compounds such as diazepam, furazolidone metabolites, malachite green, chloramphenicol, and sodium pentachlorophenate among prohibited substances. Enrofloxacin, due to its broad-spectrum antibacterial properties, excellent pharmacokinetic characteristics, low risk of cross drug resistance, and low cost, is widely used in the prevention and control of diseases in livestock, poultry farming, and aquaculture. It is one of the most frequently detected veterinary drugs in aquatic products [47,48]. The Ministry of Agriculture and Rural Affairs of China has set the maximum residue limit of enrofloxacin in fish meat at 100 μg/kg [49]. Research indicates that enrofloxacin exhibits widespread and persistent presence in aquaculture environments. In American Shad farming systems, detection rates of enrofloxacin and its metabolite ciprofloxacin reached 100% in both water and sediment samples, with concentrations peaking during autumn [50]. Residues can transform into high-risk byproducts. In tilapia experiments, enrofloxacin degrades into multiple products, with quantities varying by fish species. Some byproducts may be more harmful than the parent compound, yet traditional monitoring focuses solely on parent drug residues, underestimating combined toxicity [18]. This risk is further modulated by intraspecies variation and environmental factors. For example, enrofloxacin clearance in largemouth bass depends on water temperature, necessitating the dynamic adjustment of withdrawal periods based on temperature. Uniform standards may lead to uncontrolled residues [51]. The threat posed by enrofloxacin residues and degradation products to aquatic product quality and safety cannot be overlooked. Diazepam [52], chloramphenicol [53], malachite green [20], sodium pentachlorophenate [54], and nitrofuran compounds [55] are drugs that have been banned from detection in aquatic products, but illegal use still exists, resulting in the presence of residues or even excessive levels of harmful substances in aquatic products, posing health risks. To reduce the occurrence of veterinary drug residues and non-compliance situations for prohibited drugs in aquatic products, enterprises need to optimize farming management by maintaining water quality, ensuring reasonable feeding practices, and implementing disease prevention strategies, thus reducing drug use, and strictly adhering to withdrawal periods [56]. Governments can establish cross-provincial and even cross-border traceability mechanisms to enhance information sharing between production and consumption areas, thereby preventing regulatory gaps. Promoting aquatic product traceability platforms that document the entire process of medication use in aquaculture, transportation, and sales, will facilitate the rapid recall of problematic products.

Total volatile basic nitrogen should be the primary focus among quality indicators, as it serves as the principal measure of the freshness of aquatic products [57], enabling the timely determination of whether aquatic products have been contaminated during transportation, food service, and other stages. It is commonly used as an important monitoring indicator for the freshness and safety of aquatic products, and changes in its concentration can objectively reflect the spoilage and deterioration process of aquatic products [58,59]. To ensure the freshness, quality, and safety of aquatic products, quality control should be implemented across the entire industry chain. At the raw material stage, a stable and reliable supplier selection mechanism should be established, along with rapid detection of total volatile basic nitrogen during fishing and aquaculture stages to ensure the content does not exceed the standard. At the processing stage, optimizing processing techniques (such as pre-cooling treatment [60,61] and mild processing methods [62]) prioritized to minimize to shorten processing time. Technologies such as modified atmosphere packaging [63] and biological preservation [64] can be adopted to delay spoilage. The logistics end can enhance the cold chain delivery system and equip it with temperature and humidity monitoring devices to ensure temperature control throughout the entire process from catch to retail.

Sulfur dioxide warrants particular attention among food additives. It effectively inhibits microbial growth and delays food spoilage. Some enterprises, to improve the color of aquatic products and extend their shelf life, use excessive amounts of additives for bleaching or preservation, which may lead to residues that pose potential health risks. Long-term excessive intake can cause damage to the digestive system and liver function [65]. Therefore, it is necessary to strictly control the amount and residue standards of such additives in aquatic products. Through the synergistic effect of technological alternatives, regulatory reinforcement, and consumer education, the risks posed by sulfur dioxide and other additives can be systematically reduced, thus ensuring the safety of aquatic products and the sustainable development of the industry.

The sources of hazardous substances in aquatic products can be systematically categorized into two major types: natural sources and anthropogenic sources. Accurately identifying the source types of hazardous substances in aquatic products and quantitatively analyzing their contributions are key prerequisites for implementing differentiated risk management and source control. Heavy metal contamination primarily results from natural and anthropogenic sources [46], entering aquatic environments through pathways such as natural geological weathering, mining and smelting, agricultural runoff, domestic sewage discharge, landfill leachate, storm runoff, shipping activities, port operations, and atmospheric deposition [66,67]. In contrast, veterinary drugs, banned substances, and additives primarily originate from anthropogenic pollution, i.e., pollution sources closely linked to industrial discharges, agricultural practices, and aquaculture medication use. These pollutants traverse multiple environmental media before ultimately entering water bodies, where they accumulate in aquatic products. Therefore, integrating environmental monitoring data with operational records throughout the entire aquaculture process to quantify the contribution rates of natural background conditions and various human activities will significantly assist regulatory authorities in clarifying responsibility boundaries [68]. This data-driven approach enables management measures to shift focus from broad environmental protection to precisely target controllable human sources. By designing and implementing efficient interventions, we can effectively reduce aquatic product safety risks while safeguarding supply chain sustainability and resilience.

### 4.2. Risk Variations Across Different Stages and Their Spatiotemporal Distribution Patterns

Risks associated with aquatic products differ between the distribution stage and catering stage, with distribution posing higher risks than catering. In the distribution stage, risks are primarily concentrated in farmers’ markets and supermarkets, where common issues include “failure of cold chain management” and “contamination from multiple-party handling”. Breaks in the cold chain or temperature fluctuations directly accelerate the spoilage of aquatic products and promote microbial growth, serving as key drivers of high risks in the distribution process. Aquatic products require low temperatures throughout the entire process. However, in farmers’ markets, temperature fluctuations often occur in the seafood section due to the untimely replenishment of ice packs and the frequent opening and closing of refrigerated display cases. Moreover, supermarkets face the risk of damaged packaging during multi-tiered transportation: “production site → distribution center → storefront”. Both scenarios accelerate microbial proliferation and protein degradation, leading to exceedance indicators such as volatile basic nitrogen and total bacterial count. Risks in the catering stage are primarily concentrated in medium-sized and small restaurants. The core reason for the overall low risk in this stage is the immediate sensory feedback from consumers. The freshness of aquatic products can be quickly assessed through appearance, smell, and texture [69]. Consumers reject any abnormal products they detect, which pressures food service businesses to promptly remove non-compliant items. However, risks in the catering stage remain significant and should not be overlooked. Regulatory authorities should focus on key sampling locations such as farmers’ markets, supermarkets, and small-to-medium-sized restaurants to reduce the risks associated with aquatic products.

In terms of time, aquatic product risks exhibit significant seasonal variations, with the risk values from August to November being significantly higher than those from December to April. The underlying reason is that the high temperatures and humidity of summer not only directly promote microbial metabolism and reproduction but also accelerate endogenous protein degradation by increasing the enzymatic activity within aquatic products themselves. The synergistic effect of these two factors significantly heightens safety risks. The high temperatures and humidity in summer provide optimal conditions for the proliferation of microorganisms, accelerating protein degradation. Conversely, the low temperatures in winter inhibit microbial activity and slow down the migration of contaminants, thereby reducing risk. Regulatory authorities should optimize sampling strategies for the summer and autumn, enhance the coverage of cold chain transportation during summer, and promote technologies such as chilled packaging to extend the shelf life of aquatic products.

In spatial distribution, the risks exhibit a pattern of being higher in eastern regions than western regions and higher in coastal areas than inland areas. Zhejiang, Chongqing, and Jiangxi are classified as high-risk provinces. These areas are either major aquaculture production zones (characterized by high farming density and frequent disease outbreaks) or distribution hubs (involving multiple distribution stages and significant cold chain pressure). It is noteworthy that the flow of aquatic products within and between provinces can redistribute and amplify exposure risks to certain pollutants. Through seafood trade, contaminants can spread from high-risk areas to low-risk areas, altering the original pollution patterns and thereby creating new high-risk zones. This is closely related to social factors such as regional economic development levels, consumer market demand, and logistics network density. Strengthened sampling and inspection efforts should be implemented in high-risk provinces. The distribution of non-compliant heavy metals indicates that northern regions and coastal provinces should be given particular attention. This may be closely related to industrial emissions in coastal areas, port transportation, and sediment contamination in offshore regions, as well as the elevated baseline values characteristic of certain northern regions. The distribution of non-compliant quality indicators and additives shows scattered occurrences without discernible spatial patterns. This indicates that such risks are more closely associated with the standardization of localized, random production and storage/transportation operations. The distribution of non-compliant veterinary drugs and prohibited substances across provinces indicates that coastal and southwestern regions warrant particular attention. The high-density, intensive farming practices prevalent in coastal areas likely contribute to frequent non-compliance with veterinary drug standards. In southwestern regions, traditional farming practices, climate-related disease outbreaks, and challenges in regulatory oversight may be contributing factors. Enhanced supervision and sampling inspections are necessary to mitigate risks. The global spatial autocorrelation analysis reveals that only cadmium, enrofloxacin, furazolidone metabolites, and chloramphenicol have clustered distribution patterns, while other risk substances show random distribution. The local spatial autocorrelation analysis suggests that inspection efforts for cadmium should be strengthened in Beijing, Tianjin, Hebei, Inner Mongolia, Liaoning, Jilin, Heilongjiang, Shandong, and their surrounding provinces. These provinces, as legacy industrial bases or areas with dense concentrations of industrial and mining enterprises, may see historically accumulated heavy metal contamination in soil and water bodies enter aquaculture environments through irrigation or discharge. It is also recommended that we should strengthen the sampling intensity of enrofloxacin in Shanghai, Jiangsu, Zhejiang, Anhui, Fujian, Jiangxi, Hubei, and Hunan, as well as their neighboring provinces. These provinces constitute China’s primary aquaculture regions, characterized by diverse species and high stocking densities. As a commonly used fish medication, the spatial clustering of enrofloxacin directly reflects the disease prevention pressures and drug dependency associated with intensive aquaculture practices. Sampling intensity for furazolidone metabolites should be diminished in Hebei, Inner Mongolia, Jilin, and their neighboring provinces. This indicates that historical abuse issues in the region may have been effectively curbed through sustained oversight, with risks showing a dissipating trend. While we should increase the sampling intensity for chloramphenicol in Guangdong, Hainan, and their neighboring provinces. This may be related to some local farmers’ illegal use of banned drugs to combat diseases prone to occur in high-temperature and high-humidity environments and may also be influenced by the lingering effects of traditional, outdated prevention practices for specific livestock breeds. These characteristics provide precise targets for “regional, pollutant-specific regulation”. This reveals distinct driving mechanisms behind the spatial patterns of different hazardous substances: socioeconomic and environmental factors, such as industrial geographic layout, shifts in farming practices, variations in regulatory effectiveness, and regional economic development pathways, collectively shape the macro-geographic characteristics of risk.

### 4.3. Comparative Analysis of Research Frameworks

The three-dimensional assessment framework (“time–space–contaminant”) developed in this study represents a significant improvement over the existing research in terms of methodological objectivity, dimensional completeness, and application precision. Traditional studies predominantly employ subjective weighting methods. For instance, Li et al. [70] utilized the Analytic Hierarchy Process (AHP) in aquatic animal risk assessment, relying on expert experience to determine indicator weights, which renders the results susceptible to subjective interference. In contrast, this study employs the entropy weighting method, objectively assigning weights based on information entropy derived from 1.04 million batches of sampling data. Similar to the approach of Zhang et al. [71] to supply chain risk assessment, this method effectively mitigates subjective bias, ensuring more scientifically grounded weight determination. This provides a reliable foundation for subsequent TOPSIS model calculations, further enabling dynamic ranking and classification of risk substances to better align results with practical regulatory needs. Regarding risk assessment dimensions, most studies focus on single pollutant types or static spatial analysis, neglecting temporal dynamics and cross-regional risk transmission. This study innovatively incorporates monthly risk adjustment factors and provincial spatial risk adjustment factors. By integrating global and local spatial autocorrelation analysis, it comprehensively examines the spatiotemporal characteristics of risk substances. This reveals seasonal patterns where aquatic product risks are higher in summer and autumn and lower in winter and spring, along with a spatial clustering pattern of “higher in the east and lower in the west.” This further achieves a coupled analysis of “time–region–pollutant.” Regarding regulatory application orientation, existing studies often remain at the risk identification level. For instance, Lao et al. [40] established detection methods for heavy metals in aquatic products and health risk assessment models but did not propose targeted regulatory strategies. In contrast, this study generates high-risk “time–area” heatmaps based on a three-dimensional framework, identifying key regulatory provinces such as Zhejiang and Chongqing, as well as critical control periods from August to November, providing precise targets for targeted regulation. This aligns with the “multi-criteria decision support regulation” concept emphasized by Puertas et al. [30], yet places greater emphasis on data-driven dynamic adjustment capabilities. It better meets the current need for food safety to transition from experience-driven to data-driven approaches, providing a more systematic analytical method for aquatic product risk assessment and supporting the dynamic optimization of regulatory strategies.

### 4.4. The Importance and Value of Risk Classification and Spatiotemporal Analysis Models in Risk Prevention and Control

In aquatic product risk prevention and control systems, the integrated application of risk classification and spatiotemporal analysis models not only serves as the core technological support for enhancing risk identification accuracy but also demonstrates significant value in cost optimization, system scalability, regulatory adaptability, and practical feasibility. This study employs the entropy-weighted TOPSIS method to objectively assign weights and classify risks across 1.04 million batches of sampling data. This enables the targeted allocation of regulatory resources. Compared to traditional “one-size-fits-all” sampling approaches, it concentrates monitoring costs on high-risk areas and critical time periods, significantly reducing ineffective investments in low-risk zones. From the perspective of risk prevention, it may also justify the prioritization of supervisory resources. This cost-optimization logic aligns closely with the “spatiotemporal targeted resource allocation” concept proposed by Liao et al. in their dairy product risk grading study, enabling practical application in cost control. The “time–space–contaminant” three-dimensional framework developed in this study possesses dynamic data integration capabilities, accommodating newly identified risk substances and multi-source data dimensions. The spatiotemporal risk heatmaps and spatial autocorrelation results derived from the model analysis provide data support for formulating differentiated regulatory policies, enabling risk-driven oversight. Specifically, for high-risk cadmium accumulation zones, the model suggests combined policies of heavy metal traceability in aquaculture water bodies and targeted sampling in distribution channels. For high-risk areas regarding enrofloxacin, enhanced verification of medication records during the farming process and stricter monitoring of withdrawal periods can be implemented. This differentiated strategy significantly improves the precision and effectiveness of regulatory measures. In practice, the model’s computational logic and data requirements align seamlessly with existing regulatory frameworks. The data inputs rely solely on publicly available sampling data from market supervision authorities, eliminating the need for additional costly data collection networks. This fulfills the practical standard of a low-cost, actionable assessment model. In summary, risk grading and spatiotemporal analysis models can serve as crucial tools for advancing aquatic product risk prevention and control from “experience-driven” to “data-driven” approaches, laying a critical foundation for building an intelligent, full-chain monitoring network and a precision regulatory system.

## 5. Conclusions

Various hazards are present throughout the entire supply chain of aquatic products, encompassing aquaculture, capture, transportation, processing, and food service. However, the risk levels associated with different hazardous substances vary significantly across various stages and spatiotemporal dimensions. This study has clarified the rankings of risk substances in Chinese aquatic products and their spatiotemporal evolution patterns. While regional variations exist in the risks, the overall situation remains manageable. The top three hazardous substances identified in aquatic products are cadmium, enrofloxacin, and total volatile basic nitrogen. Among the high-risk and higher-risk substances in aquatic products across Chinese provinces, cadmium, enrofloxacin, furazolidone metabolites, and chloramphenicol have significant global spatial autocorrelation in non-compliance. Non-compliant aquatic products primarily occur in urban areas in China, with the distribution stage identified as the main process and the primary venues being farmers’ markets and supermarkets. Among the sampled provinces, Zhejiang exhibits the highest non-compliance rate. Spatiotemporal analysis indicates that aquatic products pose higher risks during the summer and autumn months, while risks are comparatively lower in winter and spring. Based on this risk classification and spatiotemporal analysis, future monitoring strategies should be optimized to implement targeted and differentiated inspection of aquatic products. This approach will enhance the efficiency of regulatory oversight and mitigate the quality and safety risks associated with aquatic products. Furthermore, monitoring frequency should be intensified in key high-risk provinces such as Zhejiang, particularly during the summer and autumn, prioritizing critical venues including supermarkets and farmers’ markets. For major risk substances such as cadmium and enrofloxacin, a comprehensive traceability system covering the entire supply chain—from aquaculture sources to point of sale—should be established. Strengthening cross-departmental collaborative oversight will facilitate the efficiency of risk warning and response mechanisms. To strengthen comprehensive control across the entire global supply chain, customs authorities should implement differentiated sampling inspections for high-risk product categories and origins, leveraging digital technologies to enhance the precision of port inspections. Furthermore, there is a need to promote the establishment of unified cross-border traceability standards to ensure information traceability throughout the product circulation process from exporting to importing countries. We should, by deepening international regulatory collaboration, construct a transnational risk-information-sharing mechanism to promptly exchange global updates on detected risk substances in aquatic products and disease outbreak information. By jointly conducting source risk assessments, we can reduce the safety risks associated with aquatic products.

This study effectively identified the spatiotemporal distribution characteristics of aquatic product risks in China through the entropy-weighted TOPSIS method and spatial autocorrelation analysis, pinpointing high-risk regions and substances. The findings provide data support for regulatory authorities to deploy resources and formulate regional prevention strategies, aligning with the demand for precision regulation. However, this study has certain limitations. The data relies on supervised sampling inspection data, which may introduce sampling bias. The uneven geographical distribution of sample sizes may weaken the stability of spatial autocorrelation analysis results. Regulatory authorities across different provinces exhibit slight variations in sampling frequency, testing priority and reporting standards. Differences in instrument models and testing methodologies employed by various laboratories result in discrepancies in detection limits for similar pollutants. Official regulatory datasets only disclose final determinations and partial detection values, failing to provide complete raw testing data. Crucially, they lack key information from upstream production stages such as farming and processing (e.g., aquaculture water quality, feed composition, medication records) alongside environmental monitoring data. This renders it impossible to conduct dose–response analyses for pollutants. It is difficult to quantify the potential impact of missing data on risk assessment outcomes, and the inability to fully elucidate the risk transmission pathway across the entire chain from source to consumer limits in-depth interpretation of spatiotemporal patterns. Consequently, it remains difficult to fully decipher the temporal and spatial patterns of contamination. Methodologically, relying solely on detection frequency and non-compliance rates means that one will fail to adequately consider the actual toxicological risks of contaminants or synergistic effects among different risk substances. This approach also leads to the neglect of chemical-form distinctions among risk agents, as one would only be using the total content as the basis for risk assessment—potentially leading to misjudgments of toxicological risks.

Future research should integrate data across the entire seafood supply chain—from aquaculture to consumption—extend data collection periods, incorporate risk analysis for diverse seafood species, and explore synergistic mechanisms among different risk substances to better guide seafood risk prevention and control. It is important to integrate environmental monitoring data (water, sediments, feed, weather, etc.) to establish multi-medium models, and deeply investigate pollutant migration and transformation patterns and their impact on aquatic product risks; we must refine the risk assessment system by incorporating morphological analysis, bioavailability, and in vivo metabolism estimates to enhance assessment accuracy; additionally, we must utilize machine learning and other methods to uncover potential patterns in big data, enabling dynamic risk prediction and providing more forward-looking support for regulatory decision making.

## Figures and Tables

**Figure 1 foods-14-04263-f001:**
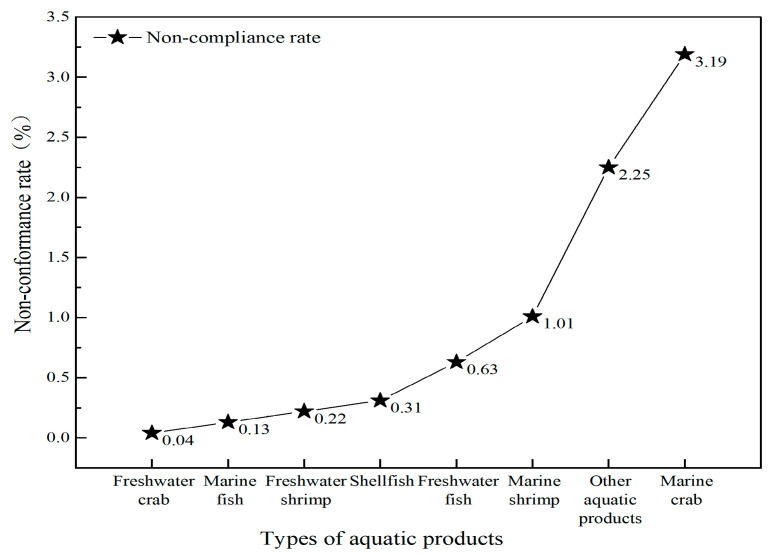
Non-compliance of aquatic product types, 2021–2023.

**Figure 2 foods-14-04263-f002:**
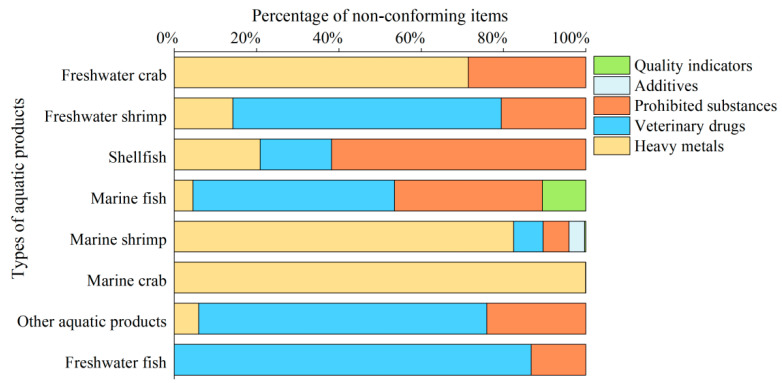
Percentage of each test item in different types of aquatic products by category.

**Figure 3 foods-14-04263-f003:**
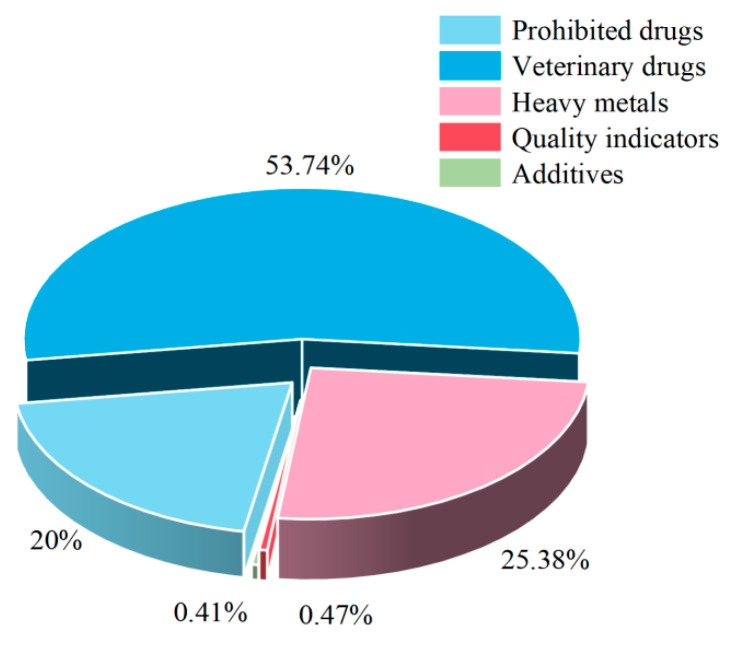
Distribution of non-compliance items of aquatic products by category.

**Figure 4 foods-14-04263-f004:**
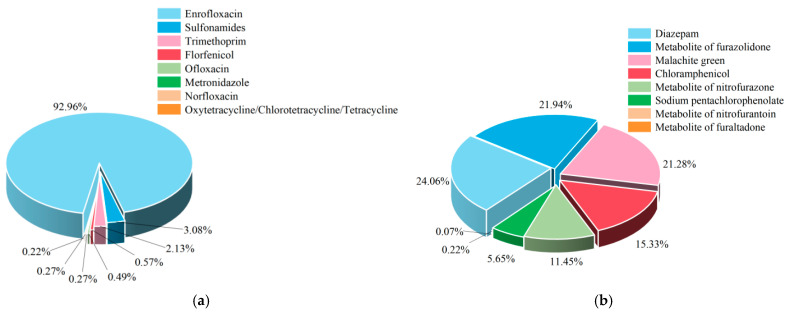
Non-compliance of inspection tests for veterinary drugs (**a**) and prohibited drugs (**b**).

**Figure 5 foods-14-04263-f005:**
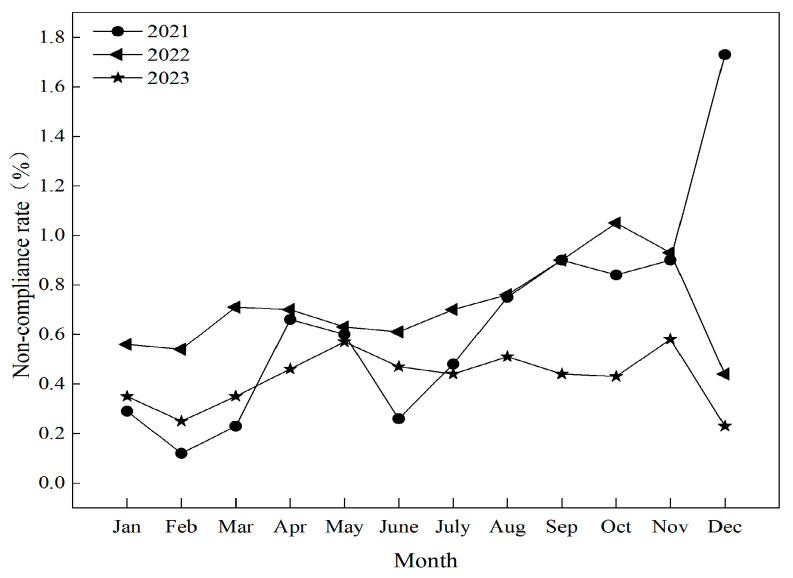
Changes in the non-compliance rate of aquatic products by month.

**Figure 6 foods-14-04263-f006:**
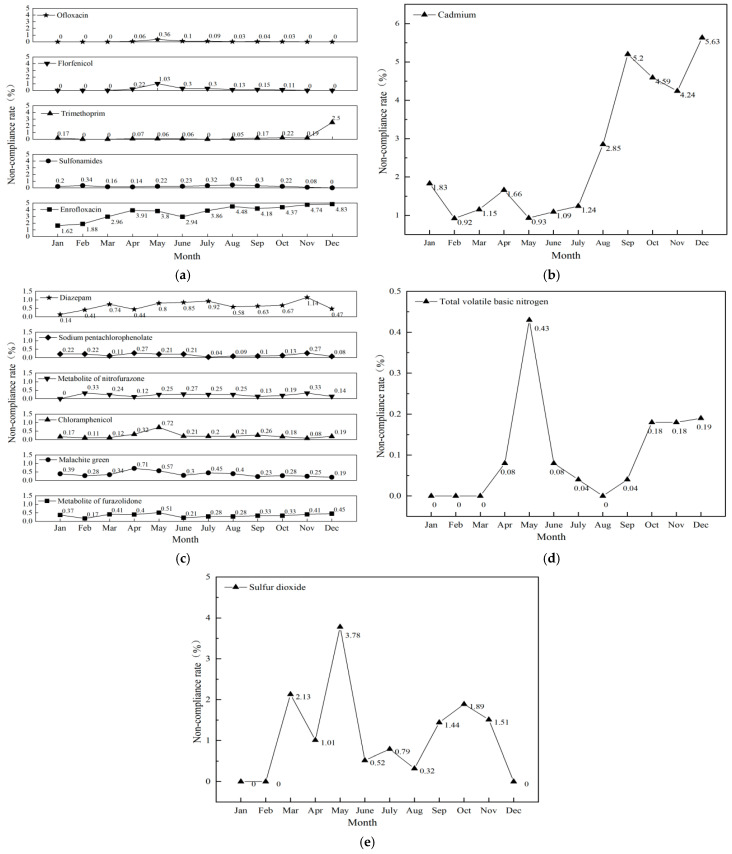
Trend of non-compliance rate of veterinary drugs (**a**), heavy metals (**b**), prohibited drugs (**c**), quality indicators (**d**), and additives (**e**) in aquatic products.

**Figure 7 foods-14-04263-f007:**
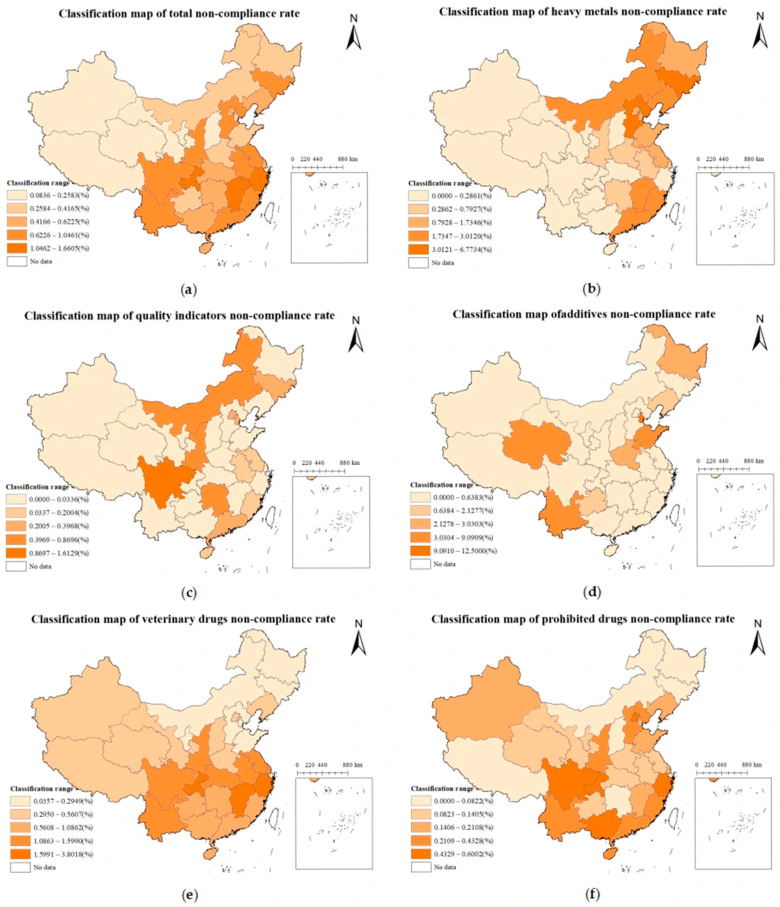
Total non-compliance rate (**a**) and non-compliance rates of heavy metals (**b**), quality indicators (**c**), additives (**d**), veterinary drugs (**e**), and prohibited drugs (**f**) in aquatic products across province in China.

**Figure 8 foods-14-04263-f008:**
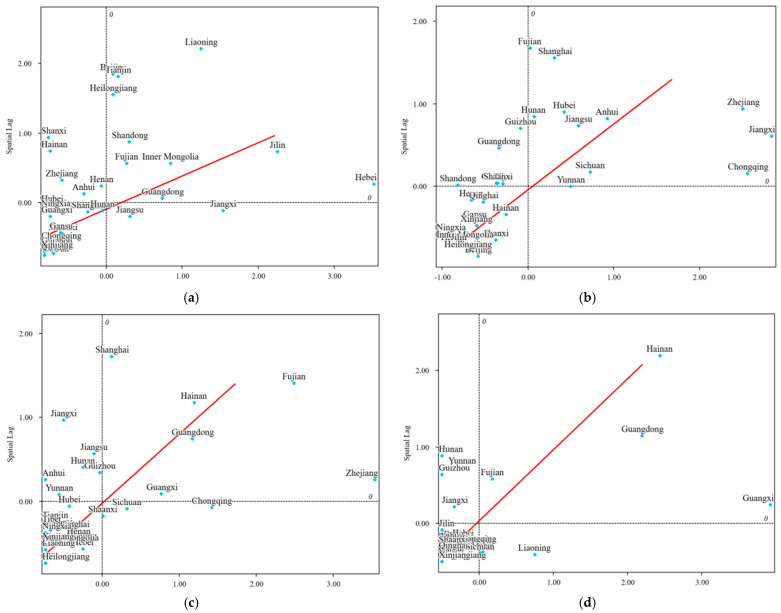
Moran scatter plot of cadmium (**a**), enrofloxacin (**b**), furazolidone metabolites (**c**), and chloramphenicol (**d**) in aquatic products. Note: In the figures, Quadrant 1 denotes positive spatial autocorrelation, representing high–high clustering of non-compliance rates; Quadrant 2 indicates negative spatial autocorrelation, representing low-high clustering; Quadrant 3 corresponds to positive spatial autocorrelation, representing low-low clustering; Quadrant 4 signifies negative spatial autocorrelation, representing high–low clustering.

**Figure 9 foods-14-04263-f009:**
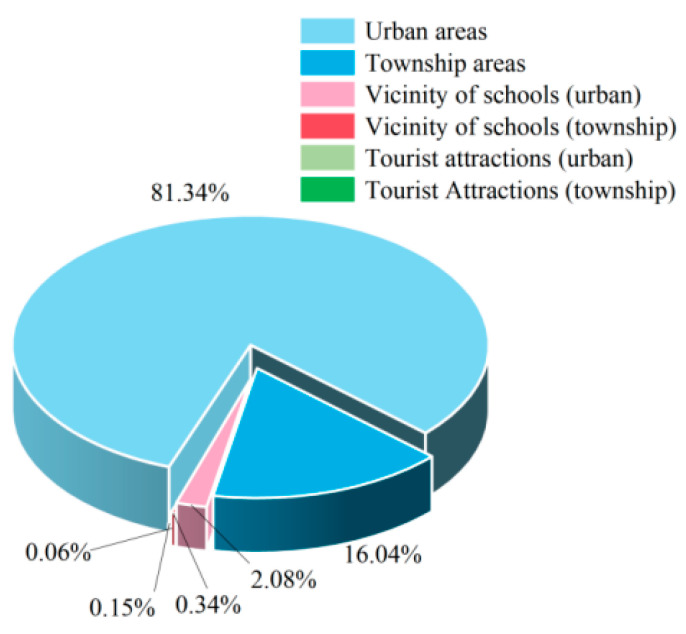
Types of sampling locations for non-compliant aquatic products in China from 2021 to 2023.

**Figure 10 foods-14-04263-f010:**
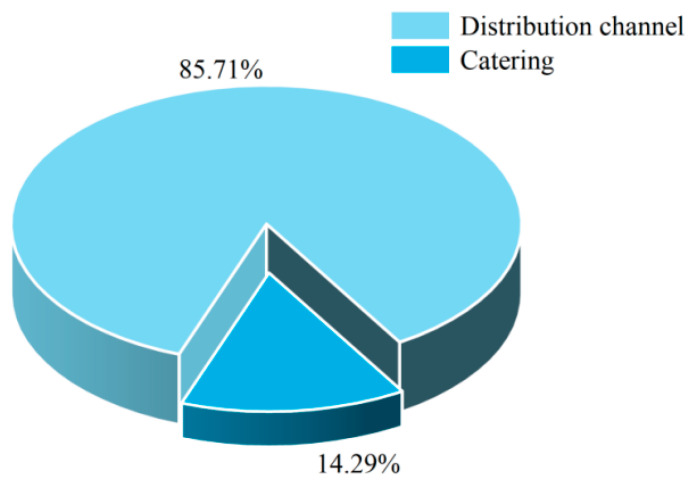
Sampling stages of non-compliant aquatic products, 2021–2023.

**Figure 11 foods-14-04263-f011:**
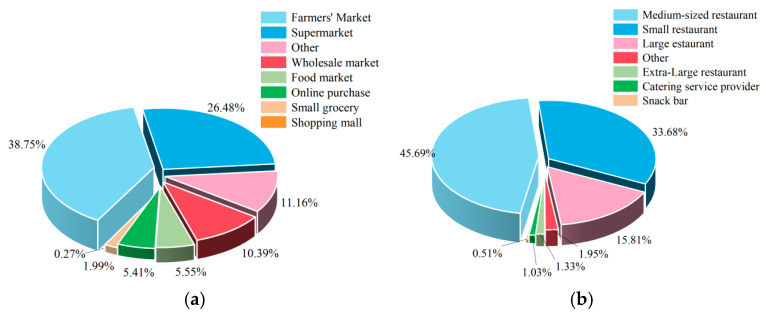
Sampling places for distribution (**a**) and catering (**b**) of unqualified products in aquatic products sampling and testing (2021–2023).

**Figure 12 foods-14-04263-f012:**
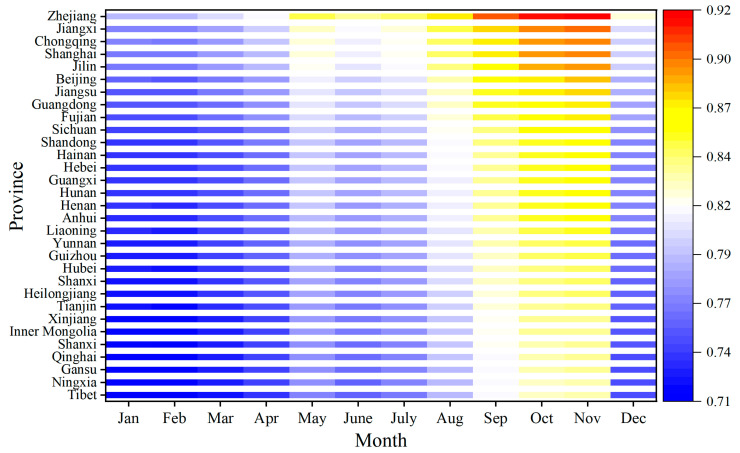
Heatmap of changes in aquatic products risk in different provinces and months. Note: In the figure, values ranging from 0.71 to 0.92 indicate low–high risk.

**Table 1 foods-14-04263-t001:** TOPSIS evaluation calculation results.

Hazardous Substance	Distribution		Catering	
	$C_{i}$	Sorting	$C_{i}$	Sorting
Cadmium	0.595	1	0.707	1
Enrofloxacin	0.579	2	0.689	2
Total volatile basic nitrogen	0.45	3	0.429	3
Sulfur dioxide	0.333	4	0.092	31
Diazepam	0.291	5	0.155	10
Methylmercury	0.274	6	0.308	4
Malachite green	0.261	7	0.154	11
Furazolidone metabolites	0.26	8	0.167	9
Chloramphenicol	0.256	9	0.147	12
Nitrofurazone metabolites	0.253	10	0.171	8
Sodium Pentachlorophenate	0.249	11	0.237	5
Metronidazole	0.248	12	0.141	13
Nitrofurantoin metabolites	0.247	13	0.139	16
Furaltadone metabolites	0.247	14	0.139	15
Sarafloxacin	0.243	15	0.13	17
Ofloxacin	0.243	16	0.123	18
Polychlorinated Biphenyls	0.239	17	0.104	27
Danofloxacin	0.238	18	0.118	23
Flumequine	0.231	19	0.118	24
Difloxacin	0.228	20	0.123	20
Oxolinic acid	0.205	21	0.123	19
Inorganic arsenic	0.2	22	0.213	6
Chromium	0.186	23	0.178	7
Histamine	0.15	24	0.088	32
Lead	0.113	25	0.14	14
Deltamethrin	0.105	26	0.104	26
Cypermethrin	0.094	27	0.095	28
Trimethoprim	0.065	28	0.045	33
Florfenicol	0.065	29	0.094	29
Sulfonamides (total)	0.052	30	0.036	34
Pefloxacin	0.045	31	0.123	21
Oxytetracycline/chlortetracycline/tetracycline (sum)	0.035	32	0.093	30
Norfloxacin	0.024	33	0.123	22
Lomefloxacin	0.023	34	0.118	25

**Table 2 foods-14-04263-t002:** Scope of risk classification.

Risk Level	Value Range	Grading
[0, 10%)	[0, 0.0513)	Lowest
[10%, 40%)	[0.0513, 0.1272)	Lower
[40%, 70%)	[0.1272, 0.2402)	Medium
[70%, 90%)	[0.2402, 0.3426)	Higher
[90%, 100%]	[0.3426, 0.7070]	Highest

**Table 3 foods-14-04263-t003:** (**a**) Risk classification of prohibited substances in aquatic products; (**b**) Risk classification of veterinary drugs in aquatic products; (**c**) Risk classification of additives, organic pollutants, and quality indicators in aquatic products.

**(a)**			
**Category**	**Hazardous Substances**	**Grading**	
		**Distribution**	**Catering**
Prohibited drugs	Chloramphenicol	Higher	Medium
	Furazolidone metabolites	Higher	Medium
	Malachite Green	Higher	Medium
	Nitrofurazone metabolites	Higher	Medium
	Sodium pentachlorophenate	Higher	Higher
	Nitrofurantoin metabolites	Higher	Medium
	Diazepam	Higher	Medium
	Furaltadone metabolites	Higher	Medium
**(b)**			
**Category**	**Hazardous Substances**	**Grading**	
		**Distribution**	**Catering**
Veterinary drugs	Enrofloxacin	Highest	Highest
	Sulfonamides (Total)	Lower	Lowest
	Trimethoprim	Lower	Lowest
	Florfenicol	Lower	Lower
	Oxytetracycline/chlortetracycline/tetracycline (Sum)	Lowest	Lower
	Metronidazole	Higher	Medium
	Ofloxacin	Higher	Lower
	Norfloxacin	Lowest	Lower
	Pefloxacin	Lowest	Lower
	Lomefloxacin	Lowest	Lower
	Deltamethrin	Lower	Lower
	Cypermethrin	Lower	Lower
	Difloxacin	Medium	Lower
	Oxolinic acid	Medium	Lower
	Flumequine	Medium	Lower
	Danofloxacin	Medium	Lower
	Sarafloxacin	Higher	Medium
**(c)**			
**Category**	**Hazardous Substances**	**Grading**	
		**Distribution**	**Catering**
Additives	Sulfur dioxide	Higher	Lower
Organic pollutants	Polychlorinated biphenyls	Medium	Lower
Quality indicators	Total volatile basic nitrogen	Highest	Highest
	Histamine	Medium	Lower
Heavy metals	Cadmium	Highest	Highest
	Chromium	Medium	Medium
	Lead	Lower	Medium
	Inorganic arsenic	Medium	Medium
	Methylmercury	Higher	Higher

**Table 4 foods-14-04263-t004:** Non-compliance items detected in different types of aquatic products.

Type of Aquatic Products	Percentage of Major Non-Compliant Items
Seawater crab	Cadmium (99.90%), chloramphenicol (0.10%)
Other aquatic products	Enrofloxacin (67.83%), furazolidone metabolites (11.43%), nitrofurazone metabolites (10.67%), cadmium (5.97%)
Seawater shrimp	Cadmium (82.47%), enrofloxacin (6.16%), furazolidone metabolites (5.34%)
Freshwater fish	Enrofloxacin (71.35%), diazepam (9.91%), malachite green (7.80%)
Shellfish	Chloramphenicol (52.44%), cadmium (20.89%), florfenicol (8.44%), enrofloxacin (8.44%)
Freshwater shrimp	Enrofloxacin (62.50%), cadmium (14.29%), furazolidone metabolites (9.82%), nitrofurazone metabolites (7.14%)
Seawater fish	Enrofloxacin (40.49%), furazolidone metabolites (16.20%), total volatile basic nitrogen (9.86%), chloramphenicol (7.75%), malachite green (7.39%), sulfonamides (total) (5.63%)
Freshwater crab	Cadmium (71.43%), nitrofurazone metabolites (14.29%), furazolidone metabolites (14.29%).

**Table 5 foods-14-04263-t005:** Global spatial autocorrelation analysis of non-compliance rates for high-risk substances in aquatic products in China (2021–2023).

Risk Substance	Moran’s I	Z Value	*p* Value
Cadmium	0.312	2.918	0.002
Enrofloxacin	0.386	3.538	0.000
Total volatile basic nitrogen	−0.155	−1.024	0.153
Sulfur dioxide	−0.124	−0.762	0.223
Diazepam	0.141	1.473	0.07
Malachite green	0.084	0.986	0.162
Furazolidone metabolites	0.319	2.971	0.001
Chloramphenicol	0.331	3.076	0.001
Nitrofurazone metabolites	0.108	1.191	0.117
Sodium pentachlorophenate	−0.04	−0.054	0.479
Metronidazole	0.002	0.301	0.382
Ofloxacin	0.229	2.212	0.013

**Table 6 foods-14-04263-t006:** Local spatial autocorrelation analysis of cadmium, enrofloxacin, furazolidone metabolites, and chloramphenicol.

Region	Cadmium		Enrofloxacin		Furazolidone Metabolites		Chloramphenicol	
	Local Moran’s I	*p* Value	Local Moran’s I	*p* Value	Local Moran’s I	*p* Value	Local Moran’s I	*p* Value
Beijing	0.168	0.003	0.476	0.058	0.115	0.113	0.068	0.184
Tianjin	0.282	0.003	0.596	0.078	0.174	0.161	0.064	0.179
Hebei	0.895	0.046	0.534	0.011	0.136	0.022	0.046	0.136
Shanxi	−0.687	0.024	0.234	0.048	0.168	0.114	0.198	0.081
Inner Mongolia	0.463	0.019	0.494	0.009	0.367	0.023	0.152	0.089
Liaoning	2.672	0.000	0.494	0.041	0.408	0.069	−0.296	0.113
Jilin	1.594	0.026	0.546	0.053	0.524	0.042	0.042	0.205
Heilongjiang	0.142	0.002	0.529	0.064	0.524	0.071	0.246	0.105
Shanghai	0.031	0.22	0.468	0.008	0.211	0.009	0.098	0.171
Jiangsu	−0.061	0.146	0.417	0.032	−0.055	0.046	0.124	0.086
Zhejiang	−0.179	0.133	2.282	0.002	0.879	0.034	0.036	0.101
Anhui	−0.035	0.202	0.73	0.004	−0.185	0.103	0.123	0.097
Fujian	0.149	0.069	0.04	0.001	3.39	0.000	0.099	0.053
Jiangxi	−0.17	0.199	1.659	0.004	−0.466	0.001	−0.068	0.128
Shandong	0.258	0.026	−0.007	0.234	0.223	0.115	0.116	0.074
Henan	−0.014	0.154	0.107	0.148	0.126	0.058	0.129	0.064
Hubei	0.031	0.196	0.369	0.005	0.023	0.223	−0.018	0.055
Hunan	0.005	0.168	0.057	0.008	−0.095	0.058	−0.43	0.001
Guangdong	0.042	0.243	−0.153	0.072	0.847	0.015	2.42	0.000
Guangxi	0.141	0.131	−0.013	0.25	0.065	0.201	0.91	0.07
Hainan	−0.526	0.08	0.083	0.176	1.364	0.026	5.168	0.005
Chongqing	0.455	0.024	0.388	0.138	−0.099	0.249	0.011	0.112
Sichuan	0.561	0.007	0.12	0.155	−0.028	0.213	−0.005	0.047
Guizhou	0.419	0.017	−0.06	0.029	−0.01	0.099	−0.309	0.033
Yunnan	0.599	0.016	−0.002	0.241	−0.044	0.22	−0.16	0.032
Tibet	0.597	0.017	−0.008	0.249	0.208	0.141	0.149	0.114
Shaanxi	0.254	0.032	−0.011	0.221	−0.002	0.174	0.146	0.068
Gansu	0.251	0.047	0.254	0.062	0.131	0.099	0.204	0.052
Qinghai	0.491	0.02	0.098	0.162	0.118	0.133	0.183	0.092
Ningxia	0.07	0.197	0.579	0.067	0.266	0.143	0.246	0.091
Xinjiang	0.479	0.041	0.275	0.093	0.278	0.11	0.246	0.091

**Table 7 foods-14-04263-t007:** Final spatiotemporal risk values for aquatic products.

Region	Month											
	Jan	Feb	Mar	Apr	May	June	July	Aug	Sep	Oct	Nov	Dec
Shanghai	0.765	0.764	0.777	0.790	0.823	0.809	0.819	0.845	0.871	0.887	0.891	0.798
Henan	0.736	0.735	0.747	0.760	0.792	0.778	0.787	0.813	0.838	0.853	0.858	0.768
Beijing	0.753	0.752	0.765	0.778	0.811	0.797	0.806	0.832	0.858	0.873	0.878	0.786
Shandong	0.737	0.737	0.749	0.762	0.794	0.780	0.789	0.814	0.840	0.855	0.859	0.769
Jiangsu	0.751	0.750	0.763	0.776	0.808	0.794	0.803	0.829	0.855	0.870	0.875	0.783
Jilin	0.763	0.762	0.775	0.788	0.821	0.807	0.816	0.843	0.869	0.884	0.889	0.796
Hunan	0.736	0.736	0.748	0.761	0.793	0.779	0.788	0.813	0.839	0.854	0.858	0.768
Guangxi	0.736	0.736	0.748	0.761	0.793	0.779	0.788	0.813	0.839	0.854	0.858	0.769
Guangdong	0.749	0.749	0.761	0.774	0.807	0.793	0.802	0.828	0.854	0.869	0.873	0.782
Zhejiang	0.790	0.789	0.802	0.816	0.850	0.836	0.845	0.872	0.900	0.916	0.921	0.824
Hainan	0.737	0.736	0.748	0.761	0.793	0.780	0.788	0.814	0.839	0.854	0.859	0.769
Fujian	0.746	0.746	0.758	0.771	0.804	0.790	0.799	0.824	0.850	0.865	0.870	0.779
Guizhou	0.729	0.728	0.740	0.753	0.785	0.771	0.780	0.805	0.830	0.845	0.849	0.760
Jiangxi	0.770	0.769	0.782	0.796	0.829	0.815	0.824	0.850	0.877	0.893	0.897	0.803
Heilongjiang	0.724	0.724	0.736	0.749	0.780	0.767	0.775	0.800	0.825	0.840	0.844	0.756
Sichuan	0.740	0.740	0.752	0.765	0.797	0.783	0.792	0.818	0.843	0.858	0.863	0.773
Chongqing	0.766	0.765	0.778	0.792	0.825	0.811	0.820	0.846	0.873	0.888	0.893	0.800
Anhui	0.735	0.734	0.747	0.760	0.791	0.778	0.787	0.812	0.837	0.852	0.857	0.767
Liaoning	0.732	0.731	0.744	0.757	0.788	0.775	0.783	0.808	0.834	0.849	0.853	0.764
Tianjin	0.723	0.722	0.734	0.747	0.778	0.765	0.773	0.798	0.823	0.838	0.842	0.754
Hubei	0.727	0.726	0.738	0.751	0.783	0.769	0.778	0.803	0.828	0.843	0.847	0.759
Yunnan	0.729	0.729	0.741	0.754	0.785	0.772	0.780	0.805	0.831	0.845	0.850	0.761
Hebei	0.737	0.736	0.748	0.761	0.793	0.779	0.788	0.814	0.839	0.854	0.859	0.769
Shanxi	0.719	0.718	0.730	0.743	0.774	0.761	0.769	0.794	0.819	0.833	0.838	0.750
Ningxia	0.716	0.715	0.727	0.740	0.771	0.757	0.766	0.791	0.815	0.830	0.834	0.747
Shaanxi	0.726	0.725	0.737	0.750	0.781	0.768	0.777	0.802	0.827	0.841	0.846	0.757
Inner Mongolia	0.720	0.719	0.731	0.744	0.775	0.762	0.771	0.795	0.820	0.835	0.839	0.751
Gansu	0.717	0.716	0.728	0.741	0.772	0.758	0.767	0.791	0.816	0.831	0.835	0.748
Qinghai	0.717	0.716	0.728	0.741	0.772	0.759	0.767	0.792	0.817	0.831	0.836	0.748
Tibet	0.715	0.714	0.726	0.739	0.769	0.756	0.765	0.789	0.814	0.829	0.833	0.746
Xinjiang	0.720	0.719	0.731	0.744	0.775	0.762	0.771	0.795	0.820	0.835	0.839	0.751

## Data Availability

The original contributions presented in the study are included in the article/Appendix A, further inquiries can be directed to the corresponding author.

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
