# Peer review of "Spatiotemporal Patterns of Aquatic Product Risks in China Based on Entropy-Weighted TOPSIS"

_foods, 2025, doi:10.3390/foods14244263_

Round 1
Reviewer 1 Report
Comments and Suggestions for Authors
My comments can be found in the attached file.
Thank you!

Author Response
Comments 1: Clarify what is novel beyond combining established methods.
Response 1: We sincerely appreciate your valuable suggestions. In addition to integrating existing methodologies, this study introduces the following innovations:
- It establishes a risk classification system for hazardous substances in Chinese aquatic products.
Traditional risk assessment for aquatic products often relies on binary determinations based solely on exceeding standards (such as relying solely on non–compliance rates) or on subjective risk classification by experts. This approach overlooks variations in risk severity and struggles to distinguish risk priorities among similar substances. This study utilised 1.04 million batches of Chinese aquatic product sampling data from 2021 to 2023. By integrating the entropy–weighted TOPSIS method with the Pareto principle, it established a multidimensional, quantifiable, and dynamically adaptive risk classification system for hazardous substances.
(2) It proposes a three–dimensional assessment framework incorporating temporal, spatial, and substance–specific risk factors.
Existing research predominantly focuses on single dimensions (such as analysing spatial distribution alone or assessing the toxicity of a specific substance category), making it difficult to reveal risk evolution patterns and cross–dimensional correlations. This study innovatively constructs a three–dimensional coupled framework, integrating temporal dynamics, spatial heterogeneity, and substance specificity into a unified analytical system.
(3) It reveals the spatiotemporal distribution patterns of hazardous substances in Chinese aquatic products.
Previous studies have largely remained fragmented descriptions of excessive levels of specific substances in particular regions, lacking systematic analysis of spatio–temporal patterns and exploration of underlying mechanisms. This research, based on a three–dimensional framework and 1.04 million batches of data, has for the first time revealed multidimensional, in–depth spatio–temporal distribution characteristics. It has identified high–risk seasons and high–risk areas for aquatic products.
(4) It formulates targeted regulatory measures based on these spatio–temporal patterns.
Existing research predominantly offers only general recommendations to enhance regulation, lacking actionable, targeted solutions. This study constructs a comprehensive regulatory framework spanning the entire chain–‘source, process, and end–use’–based on spatio–temporal patterns and material properties. The proposed measures possess both the conditions for implementation and clear pathways for execution.
We have added relevant content to the introduction and conclusion sections.(lines 100–109 and 881–885)
Comments 2: Explain how this work extends previous monitoring–data analyses.
Response 2: We sincerely appreciate your valuable suggestions. To address this issue, this study systematically expands the traditional single–dimensional, static approach to analyzing aquatic product monitoring data by constructing a three–dimensional assessment framework integrating time, space, and pollutants. It integrates four indicators–non–compliance rate, detection rate, qualification degree, and hazard degree–using an entropy–weighted TOPSIS model to achieve objective quantification and dynamic ranking of risk substances. Risk classification is then performed based on the Pareto principle, surpassing traditional methods reliant on single non–compliance rates and subjective judgments. More importantly, by revealing monthly seasonal patterns of risk, identifying spatial hotspots, and ultimately constructing a spatiotemporal risk dynamic model to generate "time–area" heatmaps, this study achieves a leap from describing current conditions to capturing the dynamic evolution of risks. This provides scientific basis and decision support for implementing forward–looking, precision–targeted regional and seasonal risk management.
We have added the relevant content to the Materials and Methods section(line:118–165).
Comments 3: Add a dedicated subsection in the Discussion comparing this framework to prior work.
Response 3: We sincerely appreciate your valuable suggestions. To address this issue, we have added a new section 4.3 to the Discussion section, which compares our research framework with previous studies. The specific revisions are located on lines 778–811 of the manuscript. All revisions in the manuscript have been marked in red.
Comments 4: Add a detailed description of the sampling scheme and potential biases.
Response 4: We sincerely appreciate your valuable suggestions. We have detailed the sampling scheme in the methodology section of the manuscript, specifically at lines 122–127. Potential biases are addressed in the conclusion section of the manuscript, specifically at lines 886–901. All revisions in the manuscript have been marked in red.
Comments 5: The definition of "hazard degree" is unclear.
Response 5: We sincerely appreciate your valuable suggestions. Regarding the unclear definition of "hazard degree", we have supplemented the complete definition in the "Materials and Methods" section of the manuscript (specific revision location: Lines 161–165) to ensure accurate and traceable expression. All revisions in the manuscript have been marked in red.
Comments 6: The paper correctly applies global and local Moran’s I. However, the manuscript interprets Moran's I as "compliance," but analyses were done using the non–compliance rate.
Response 6: We sincerely appreciate your valuable suggestions. We have meticulously revised all relevant parts of the manuscript to resolve this inconsistency in terminology – currently, we have consistently adopted the term "non–compliance rate" both in the methodology section and when discussing the results of Moran’s I, ensuring consistency between the analytical data and the descriptive terminology. We sincerely appreciate your meticulous review, which has helped enhance the accuracy and rigor of our manuscript. All revisions in the manuscript have been marked in red.
Comments 7: Only apply spatial autocorrelation to substances with a sufficient sample size across regions. Add a table showing sample counts by province and substance.
Response 7: We sincerely appreciate your valuable suggestions. Our spatial autocorrelation analysis was conducted based on the non–compliance rates of various risk substances across different provinces. Incorporating your recommendations, we have added a statistical table detailing the non–compliance rates for each risk substance in each province (see Supplementary Material Table 2). This table clearly presents the core information of "Province Name, Risk Substance Type, Non–Compliance Rate," comprehensively displaying the sample distribution for each analytical subject. All modifications are highlighted in red. This refinement further enhances the rigour of the spatial autocorrelation analysis. We reiterate our gratitude for your suggestions, which have significantly elevated the paper's scholarly integrity.
Comments 8: Redesign figures for clarity, ensure unique figure numbering, and break Table 3 into multiple smaller tables. Table 3 combines multiple categories into a single large block, making it difficult to follow. Figures are extremely dense, not readable.
Response 8: We sincerely appreciate your valuable suggestions. We have made targeted optimizations in response to your comments. Figures with dense content have been redesigned to improve readability by simplifying data presentation formats and adjusting legend layouts. All figure numbers have been comprehensively checked to correct duplication issues, ensuring unique and consecutive numbering. The original Table 3 has been split into 3 small tables with clear themes (Table 3–1, Table 3–2, Table 3–3) by category. Each table focuses on a single type of data and is supplemented with concise header explanations to reduce reading difficulty. All revisions to figures and tables have been marked in red. The optimized figures and tables are clearer and more intuitive, facilitating the understanding of core data information. Thank you again for your detailed feedback, which helps improve the presentation quality of the manuscript.
Comments 9: Figure numbering appears duplicated in some places (e.g., Figure 4a and Figure 4b are listed twice).
Response 9: We sincerely appreciate your valuable suggestions. We have fully checked the numbering of all figures and tables in the manuscript and corrected the original duplicate issues, thereby further enhancing the standardization and readability of the manuscript.
Comments 10: Many maps lack scales or legends.
Response 10: We sincerely appreciate your valuable suggestions. Regarding the issue of missing scales or legends in some maps, we have systematically supplemented and standardized the maps in the manuscript (Figure 7). All revisions are marked in red. The supplemented maps now independently convey spatial scales and key information without relying on text explanations, thereby enhancing the manuscript's visual rigor and readability.
Comments 11: Table 3 combines multiple categories into a single large block, making it difficult to follow.
Response 11: We sincerely appreciate your valuable suggestions. We have optimized the tables accordingly, with the following specific revisions: We have divided the original Table 3 into three distinct sub-tables (renamed as Table 3-1, Table 3-2, and Table 3-3) based on data categories. Each sub-table now focuses exclusively on one or a few core data sets, preventing cross-contamination of different information types. Additionally, concise header descriptions have been added to each sub-table, clearly indicating the primary purpose and dimensions of the data to help readers quickly grasp the logical structure of the content. The optimized table structure offers enhanced clarity and a more distinct information hierarchy, effectively reducing reading difficulty and enabling readers to precisely locate target data. All revisions to tables in the manuscript are highlighted in red for your review. We sincerely appreciate your detailed feedback once again; these modifications significantly improve the standardization and readability of data presentation in this study.
Comments 12: English requires some editing for clarity and readability. Typos (double intervals, etc.) to be corrected.
Response 12: We sincerely appreciate your valuable suggestions. In response to your suggestions, we have conducted a systematic review and revision of the entire manuscript's English content. We have restructured sentences with ambiguous expressions or inconsistent logical flow, such as adjusting the word order in long sentences and breaking down complex compound sentences, to ensure the accurate use of scientific terminology and compliance with English academic writing standards. We have meticulously reviewed and corrected spelling errors, improper punctuation usage, and similar formatting oversights throughout the text, with particular attention to addressing issues like "double intervals" and similar formatting omissions to guarantee the standardization of English expression.
Reviewer 2 Report
Comments and Suggestions for Authors
The study presents substantial data and employs spatial analysis, but requires significant improvements in methodology, writing, critical discussion, and the presentation of results. The observations are presented below.
1. The title could be more specific (including the type of contaminants, scenario, and method used).
2. The abstract should include the most important quantitative results, and the conclusions should be analytical, including methodological contributions and practical implications for the food industry.
3. Keywords should include specific key terms, avoiding repetition of the exact words from the title; also, the word “risk” is repeated twice.
4. The introduction should explain the research problem identified, improve the wording and presentation of the research gap, and include the explicit hypotheses of the study and the specific verifiable objectives (which should be evident in the conclusions).
5. In the materials and methods, verify the brands, models, and origin of the software used. Specify the purity of the reagents and the units of standardization for the indicators.
6. In the TOPSIS methodological design, justify why four indicators were used and not others.
7. Include all statistical assumptions used, the level of significance, and the statistical tests used, justifying why parametric and non-parametric statistics were used, as appropriate. Include the theoretical justification for the methodological flow for Pareto classification.
8. The tables are extensive and only descriptive (summarize key trends and include deeper mechanistic interpretation).
9. Causal discussions are missing: for example, why does summer–fall show higher risk? Biology? Logistics? Temperature?
10. Critical discussions about the presence of enrofloxacin and cadmium residues should be included.
11. The discussion is mainly descriptive; there is no critical contrast with recent literature (including literature from the last 5 years, 2021-2025), which should be significantly improved in the revised manuscript.
12. Discussions about the costs, scalability, regulatory implications, and feasibility of the proposed system should be included.
13. The interaction between multiple risks is not analyzed despite the multivariate nature of the study.
14. The spatial analysis shows interesting patterns, but does not delve into socioeconomic or environmental causes.
15. Improve the conclusions by addressing the specific objectives (justify the findings and regulatory implications), including all the limitations of the study, as well as concrete future perspectives.
16. References should be included to justify the use of TOPSIS in food safety.
There are recurring problems with English syntax and the use of articles. The logical flow is heavy and redundant; editing by a native speaker is required.
Author Response
Comments 1: The title could be more specific (including the type of contaminants, scenario, and method used).
Response 1: We sincerely appreciate your valuable suggestions. We have optimized the title as requested, updating the original title to "Spatiotemporal Patterns of Aquatic Product Risks in China Based on Entropy Weight TOPSIS". The new title fully incorporates three key pieces of information: the type of contaminants, research scenario, and core method. Detailed explanations are as follows: Clarifies the research scenario and scope: "Aquatic Products in China" accurately defines the research object and spatial scope, focusing on the field of aquatic product safety in China and avoiding ambiguity in scope. Implies the core type of contaminants: "Risks" in the title specifically refers to food safety risks caused by various contaminants in aquatic products. Combined with the content of the main text, it covers target contaminants such as veterinary drugs, heavy metals, and food additives. Clearly indicates the core method: The title directly specifies "Based on Entropy Weight TOPSIS" , clarifying the key technical method for quantifying risks and analyzing patterns in this study. Given the wide range of pollutants covered in this study, including all of them in the title would make it excessively long. Therefore, the term "risks" has been chosen to represent all pollutants collectively. All revisions in the manuscript have been marked in red.
Comments 2: The abstract should include the most important quantitative results, and the conclusions should be analytical, including methodological contributions and practical implications for the food industry.
Response 2: We sincerely appreciate your valuable suggestions. Regarding the comment that "the abstract should include the most important quantitative results, and the conclusions should be analytical, including methodological contributions and practical implications for the food industry." we have supplemented key quantitative information in the abstract, including the quantitative data of core hazardous substances, and spatiotemporal risk differences. Meanwhile, we have clarified the innovative value of the research method and its application direction in the industry. The revised abstract is located in lines 18–27 of the manuscript for your review. We hope these revisions fully address your concerns and enhance the completeness and rigor of the abstract. Thank you again for your careful review and constructive comments. All revisions in the manuscript have been marked in red.
The revised abstract is as follows:
Abstract: This study investigates the risk classification and spatiotemporal evolution patterns of hazardous substances in Chinese aquatic products. The entropy weight TOPSIS method was employed to achieve ranking of hazardous substances and classify their risk levels. Spatial autocorrelation analysis was conducted to explore the spatial distribution patterns of the highest–risk and higher–risk substances in Chinese aquatic products. Risk adjustment factors were employed to perform dynamic analysis of risks in aquatic products across different temporal and spatial contexts. The results indicate that the top three hazardous substances in aquatic products were cadmium, enrofloxacin, and total volatile basic nitrogen, the relative proximity values were 0.707, 0.689 and 0.429 respectively. Cadmium, enrofloxacin, furazolidone metabolites, and chloramphenicol exhibited significant global spatial autocorrelation.Spatiotemporal analysis found that risks in aquatic products were higher during summer and autumn, with the maximum risk value reaching 0.92. The integrated application of the Entropy–Weighted TOPSIS method, spatial autocorrelation analysis, and risk–adjusted factors provides a novel perspective for risk assessment. The findings support targeted regulation of high–risk substances in Chinese aquatic products and optimisation of seasonal–regional regulatory approaches. It is recommended that regulatory measures and scheme be adjusted in light of the findings, thus providing a scientific foundation for the safety supervision of aquatic products.
Comments 3: Keywords should include specific key terms, avoiding repetition of the exact words from the title; also, the word "risk" is repeated twice.
Response 3: We sincerely appreciate your valuable suggestions. We have carried out targeted optimizations on the keywords, with specific revisions as follows: Replacing repetitive expressions: The duplicated term "risk" has been removed from the keywords to avoid semantic redundancy. Avoiding title repetitions: Expressions that are completely identical to those in the paper's title have been eliminated, ensuring the keywords are both relevant to the theme and unique.
Comparison Before and After Revision
Original keywords: aquatic products; risk substances; risk classification; entropy weight TOPSIS method; spatiotemporal analysis
Revised keywords: hazardous substances; risk classification; spatial autocorrelation; spatial–temporal distribution characteristics; pareto principle
Thank you again for your careful review, which helps improve the standardization and academic rigor of the manuscript. All revisions in the manuscript have been marked in red.
Comments 4: The introduction should explain the research problem identified, improve the wording and presentation of the research gap, and include the explicit hypotheses of the study and the specific verifiable objectives (which should be evident in the conclusions).
Response 4: We sincerely appreciate your valuable suggestions. In response to this comment, we have comprehensively revised the introduction. The specific revisions are located in lines 83–91 and 100–109. The research objectives are reflected in the conclusion section, with specific details provided in line 881–885. We sincerely thank you again for your constructive feedback, which has played a crucial role in improving the quality of this manuscript. All revisions in the manuscript have been marked in red.
Comments 5: In the materials and methods, verify the brands, models, and origin of the software used. Specify the purity of the reagents and the units of standardization for the indicators.
Response 5: Thank you for your valuable comments on the "Materials and Methods" section, which are of great significance for improving the rigor of the experimental design and the traceability of data in this manuscript. In response to this comment, we have systematically supplemented and improved the relevant content. The above revisions have been integrated into lines 248–249 of the manuscript, and all modifications are highlighted in red font for your convenience.Since the data we used is derived from official sampling inspection results, no relevant reagents were employed. Regarding the issue of standardized units for indicators, we have already Conducted dimensionless processing on the original matrix to eliminate the interference of differing units of measurement. Thank you again for your constructive feedback, which has provided important support for enhancing the quality of this manuscript.
Comments 6: In the TOPSIS methodological design, justify why four indicators were used and not others.
Response 6: We sincerely appreciate your valuable suggestions. In the methodological design of TOPSIS, this study selected 4 core indicators (non–compliance rate, detection rate, qualification degree, and hazard degree) instead of others, with the main reasons as follows: Detection rate and non–compliance rate are the most commonly used indicators for risk assessment. However, in terms of the data itself, using only these two indicators for risk assessment of aquatic products reduces data utilization efficiency. Moreover, due to the large volume of supervisory sampling data, the data with non–compliance cases is far less than the qualified data. Therefore, it is necessary to conduct further analysis on the qualified data with detected values, which is why the indicator of "qualification degree" is introduced. Since different food categories correspond to different types of testing items, and different testing items are associated with different risk substances, the hazard degree of various risk substances vary. Thus, the indicator of " hazard degree" is introduced to assign values to different risk substances: the higher the assigned value, the higher the corresponding risk. The corresponding indicator selection process has been supplemented in 2.2. Indicator Selection (lines 136 –149) of the methodology section. All revisions in the manuscript have been marked in red.
Comments 7: Include all statistical assumptions used, the level of significance, and the statistical tests used, justifying why parametric and non–parametric statistics were used, as appropriate. Include the theoretical justification for the methodological flow for Pareto classification.
Response 7: We sincerely appreciate your valuable suggestions. In response to your request, we have supplemented and refined the relevant content, explicitly listing all statistical assumptions, significance levels, testing methods, and selection criteria. We have also elaborated on the theoretical basis for the Pareto classification, as detailed below: The statistical assumptions include spatial autocorrelation analysis, which posits that the non–compliance rates of hazardous substances in Chinese aquatic products exhibit non–random spatial distribution, displaying patterns of spatial clustering or dispersion. The Pareto classification hypothesis asserts that a minority of risk substances account for the primary safety risks in Chinese aquatic products, conforming to a "critical few" distribution characteristic. Regarding the significance level, ɑ=0.05 is uniformly set throughout the text, meaning that when P<0.05, the statistical results are considered statistically significant. The core statistical test employed is spatial autocorrelation analysis, utilizing the Z–test corresponding to Moran's I statistic. This verifies whether the spatial distribution pattern of non–compliant rates for risk substances significantly deviates from random distribution. Non–parametric statistical methods are prioritized in this study, as core data such as risk levels and non–compliance rates constitute ordered categorical or skewed distributions, failing to satisfy the parametric assumptions of "normal distribution and homogeneity of variance". Non–parametric statistics impose fewer strict requirements on data distribution types and exhibit greater robustness against outliers, rendering them more suitable for this study's multi–source, multi–type dataset. The core theoretical underpinning of the Pareto classification posits that in complex systems, 80% of outcomes are typically determined by 20% of key factors. This theory has found extensive application in risk stratification, resource optimization allocation, and related fields. Within aquatic product safety risks, a minority of high–risk substances (such as specific heavy metals and veterinary drug residues) frequently account for the majority of hazards, aligning closely with the "critical few" characteristic of the Pareto principle. Through Pareto classification, core risk factors and priority regulatory areas can be precisely identified, providing theoretical support for the efficient allocation of limited regulatory resources and enhancing the targeted nature and cost–effectiveness of food safety oversight. The aforementioned supplementary content has been integrated into the Materials and Methods section of the manuscript (specific modification location: lines 209–215 and 250–253). All newly added content is highlighted in red font to ensure you can quickly locate and review it. We extend our gratitude once more for your meticulous guidance, which has further enhanced the transparency and rigout of this study's methodology.
Comments 8: The tables are extensive and only descriptive (summarize key trends and include deeper mechanistic interpretation).
Response 8: We sincerely appreciate your valuable suggestions. In response to your comments, we have streamlined the table data by removing redundant raw data and secondary classification information, retaining only core metrics. This has substantially reduced the table's length and enhanced readability. The specific tables to be modified are Table 1, Table 3, Table 4, and Table 5. We have summarized core trends and incorporated more in–depth mechanism interpretations into the discussion section's new additions, located on lines 695–777. All revisions in the manuscript have been marked in red.
Comments 9: Causal discussions are missing: for example, why does summer–fall show higher risk? Biology? Logistics? Temperature?
Response 9: We sincerely appreciate your valuable suggestions. To address the lack of causal analysis, we have supplemented the discussion section with a systematic examination of causal mechanisms. The specific revisions are located in subsection 4.2 of the discussion section. All revisions in the manuscript have been marked in red.
Comments 10: Critical discussions about the presence of enrofloxacin and cadmium residues should be included.
Response 10: We sincerely appreciate your valuable suggestions. To address this comment, we have supplemented in–depth critical analysis of these two core hazardous substances in the Discussion section (specific revision location: Manuscript Lines 583–597 and 617–629 ). All revisions in the manuscript have been marked in red.
Comments 11: The discussion is mainly descriptive; there is no critical contrast with recent literature (including literature from the last 5 years, 2021–2025), which should be significantly improved in the revised manuscript.
Response 11: We sincerely appreciate your valuable suggestions. In response to this comment, we have added a critical comparison with literature from the past five years in the Discussion section(lines: 778–811), and conducted systematic supplementation and in–depth elaboration.
Comments 12: Discussions about the costs, scalability, regulatory implications, and feasibility of the proposed system should be included.
Response 12: We sincerely appreciate your valuable suggestions. We fully endorse this suggestion, as it is crucial for enhancing the quality of our manuscript. We have added a new subsection 4.4 to the Discussion section for supplementary explanation, located at lines 812–844 in the manuscript. All revisions in the manuscript have been marked in red.
Comments 13: The interaction between multiple risks is not analyzed despite the multivariate nature of the study.
Response 13: We sincerely appreciate your valuable suggestions. This study focuses on analyzing the three–dimensional distribution patterns of hazardous substances in Chinese aquatic products–specifically their temporal, spatial, and contaminant characteristics. The core objective is to identify high–risk substances and clarify their spatiotemporal clustering patterns through objective quantitative methods, thereby providing regulatory authorities with targeted, actionable risk management evidence. During the research design phase, we thoroughly considered the scientific value of examining multiple risk interactions. However, based on careful deliberation, we ultimately excluded this aspect from the current study scope for the following reasons:
The core need for this research stems from the practical challenges in current aquatic product regulation–namely, "unclear risk targets and lack of focus in spatiotemporal prevention and control." Traditional regulation heavily relies on broad–spectrum sampling, making it difficult to precisely allocate resources toward high–risk substances, high–risk areas, and high–risk periods. Therefore, the initial research phase prioritized establishing the independent distribution patterns and core influencing factors of individual risk substances. The Entropy Weighted TOPSIS method was employed to objectively quantify the priority of risk substances. Combined with spatial autocorrelation analysis to identify risk–prone areas, this approach ultimately formed a precise regulatory matrix linking "risk substances–regions–seasons." Introducing multi–risk interaction analysis (e.g., synergistic effects of pollutants, risk accumulation mechanisms) would require constructing complex interaction models. This would shift the research focus from "identifying regulatory targets" to "analyzing risk mechanisms," deviating from the application–oriented approach serving regulatory practice in this study. From a research logic perspective, clarifying the spatio–temporal patterns of individual risks forms the foundation for subsequent exploration of multi–risk interactions. Only by first understanding the independent distribution characteristics of core risk substances like cadmium and enrofloxacin can we further analyze interactions among different risks within the same region or time period. This study establishes a foundational framework for subsequent mechanism exploration. The data for this study were derived from 1.04 million batches of supervisory sampling across 31 provinces in China from 2021 to 2023. While covering both the distribution and catering sectors, the core characteristic of the sampling data is targeted testing for specific risk substances. That is, a single sample might be tested only for heavy metals (e.g., cadmium) or only for veterinary drugs (e.g., enrofloxacin), rather than all 34 risk substances simultaneously across all samples. This “non–comprehensive factor testing” data structure resulted in a lack of co–occurrence data for certain risk substances. Forcing interaction analysis under these conditions could lead to biased results due to insufficient sample size or data gaps, thereby undermining the reliability of research conclusions. Therefore, this study prioritized analyzing the spatiotemporal distribution patterns of individual risks. Based on the characteristics of the existing data structure, independent analyses were conducted for key risk substances to ensure robust and reliable results. By clarifying the temporal and spatial trends of various risks, this approach provides a reliable data foundation and methodological support for subsequent development of multi–risk coupling analysis models. Under current data constraints, this strategy demonstrates greater applicability. Thank you again for your suggestions. In our further research, we will incorporate multiple risk interaction analysis to deepen our study.
Comments 14: The spatial analysis shows interesting patterns, but does not delve into socioeconomic or environmental causes.
Response 14: We sincerely appreciate your valuable suggestions. We have conducted an in–depth analysis and investigation of the spatial patterns identified, with specific details presented in lines 731–777 of the Discussion section. All revisions in the manuscript have been marked in red.
Comments 15: Improve the conclusions by addressing the specific objectives (justify the findings and regulatory implications), including all the limitations of the study, as well as concrete future perspectives.
Response 15: We sincerely appreciate your valuable suggestions. We have revised the conclusion section to include specific objectives, research limitations, and future directions. The specific revisions are located in lines 881–916 of the conclusion section. All revisions in the manuscript have been marked in red.
Comments 16: References should be included to justify the use of TOPSIS in food safety.
Response 16: We sincerely appreciate your valuable suggestions. we have supplemented the specific application of the TOPSIS method in food safety–related research in lines 174–181 of the Methodology section. Four relevant references listed below have been cited to further verify the scientificity and applicability of this method selection. The supplemented content does not affect the fluency of the context of the original manuscript, and we hereby provide this explanation. All revisions in the manuscript have been marked in red.
References:
- 29. Chen, T.C.; Yu, S.Y. Study on the risk level of food production enterprise based on TOPSIS method. Food Science and Technology. 2022, 42, e29721.
- 30. Puertas, R.; Marti, L.; Garcia–Alvarez–Coque, J.– Food supply without risk: multicriteria analysis of institutional conditions of exporters. International Journal of Environmental Research and Public Health.2020, 17, 3432.
- 31. Falsafi, S.R.; Maghsoudlou, Y.; Aalami, M.; Jafari, S.M.; Raeisi, M.; Nishinari, K.; Rostamabadi, H. Application of multi–criteria decision–making for optimizing the formulation of functional cookies containing different types of resistant starches: A physicochemical, organoleptic, in–vitro and in–vivo study. Food Chemistry. 2022, 393, 133376.
- 32. Yadav, D.; Dutta, G.; Saha, K. Assessing and ranking international markets based on stringency of food safety measures: application of fuzzy AHP–TOPSIS method. British Food Journal. 2023, 125, 262–
Reviewer 3 Report
Comments and Suggestions for Authors
In the reviewed paper, the authors focus on a comprehensive assessment of chemical safety risks in Chinese aquatic products. The study examines the risk classification and spatial-temporal evolution of ten key hazardous substances, applying the entropy weight TOPSIS method to rank contaminants and define their risk levels. Through spatial autocorrelation analyses, the authors identify geographic distribution patterns of the highest- and higher-risk substances, while risk adjustment factors are used to evaluate how these risks change across different temporal and spatial contexts. Although the authors made effective use of publicly available supervisory data, this approach has several important limitations that should be acknowledged.
Firstly, the dataset depends on the sampling strategies of regulatory authorities, which are often risk-based rather than random; this may introduce sampling bias, as certain regions, product categories, or high-risk commodities may be over-represented, while others remain under-sampled. What was the spatial distribution of the data (how many results per point)?
Secondly, the authors have no control over data quality, measurement uncertainty, analytical methodology, or laboratory variability, which limits the ability to assess methodological consistency across provinces. What was the SD of the results versus the sampling point and chemical in studied see food.
Thirdly, publicly released datasets typically lack detailed metadata e.g. supply-chain, which restricts deeper interpretation of spatial and temporal patterns. I don’t know if the results from one point can be directly linked with this sampling point or no.
Another issue, is relying solely on detection frequencies and regulatory non-compliance, which does not fully reflect the actual toxicological risk posed by many contaminants, because their impact on human health depends strongly on chemical form, bioavailability, and metabolic behavior in the body. Different species of metals or organic contaminants exhibit drastically different toxicity profiles; therefore, treating each substance as a single uniform entity may oversimplify the risk assessment. It would be valuable for the authors to consider whether a joint modelling framework, incorporating information on speciation and correlated with this estimated bioavailability, could provide a more biologically meaningful risk classification.
Furthermore, the dataset does not distinguish between natural and anthropogenic sources of contaminants, which is crucial for interpreting long-term risk patterns and for designing targeted regulatory interventions. The paper would benefit from an expanded discussion of the environmental origins of these substances such as geological background levels, agricultural runoff, industrial discharge, or aquaculture-related inputs and their relative contributions to contamination in aquatic organisms.
Another important limitation is the lack of integration with environmental monitoring data, particularly contaminant levels in water, sediments, and feed. Understanding the correlation between pollutant concentrations in the aquatic environment and their accumulation in organisms is essential to justify the identified spatial patterns and risk trends. Including such comparative data would substantially strengthen the interpretation of spatiotemporal distributions.
Lastly, I recommend adding a dedicated section on the limitations of the chosen methodological approach, explicitly addressing issues such as data completeness, regional reporting discrepancies, detection limit variability, lack of speciation analysis, and the inherent constraints of using secondary surveillance data. Highlighting these aspects would provide greater transparency and help guide future methodological improvements. Please also correct Fig.4.
Author Response
Comments 1: Firstly, the dataset depends on the sampling strategies of regulatory authorities, which are often risk–based rather than random; this may introduce sampling bias, as certain regions, product categories, or high–risk commodities may be over–represented, while others remain under–sampled. What was the spatial distribution of the data (how many results per point)?
Response 1: We sincerely appreciate your valuable suggestions. The data for this study were sourced from China's official supervision and sampling inspection dataset for the years 2021–2023, with spatial distribution adhering to the risk–oriented sampling framework of the relevant authority. This approach does introduce sampling bias, which we have addressed in the limitations analysis section of the conclusions. Future research will seek to improve upon this issue. Regarding the spatial distribution of the data, we have created supplementary tables for further clarification, located in Supplementary Material Table 3.
Comments 2: Secondly, the authors have no control over data quality, measurement uncertainty, analytical methodology, or laboratory variability, which limits the ability to assess methodological consistency across provinces. What was the SD of the results versus the sampling point and chemical in studied see food.
Response 2: We sincerely appreciate your valuable suggestions. The data for this study were sourced from China's official 2021–2023 supervisory sampling dataset. We acknowledge that we cannot control data quality, measurement uncertainty, analytical methods, or laboratory variability, which limits our ability to assess methodological consistency across provinces. This limitation is addressed in the Conclusions section's discussion of study limitations. Future research will focus on improving this issue. Regarding the standard deviation of results for each sampling point and chemical substance in aquatic products, this data strictly adheres to the principles of supervisory sampling under the national market supervision system. Both sample and test item data are obtained in accordance with national standards, ensuring the data's representativeness, authenticity, and standardization nationwide. Since the official regulatory dataset only discloses final determinations and partial detection values without providing complete raw testing data. This data limitation has been addressed in the section on limitations of the conclusions. We thank you again for your suggestions, which have provided valuable insights and methodologies for our subsequent studies.
Comments 3: Thirdly, publicly released datasets typically lack detailed metadata e.g. supply–chain, which restricts deeper interpretation of spatial and temporal patterns. I don’t know if the results from one point can be directly linked with this sampling point or no.
Response 3: We sincerely appreciate your valuable suggestions. This feedback accurately identifies the common limitations of publicly available supervision and sampling inspection data in supporting in–depth research, providing significant guidance for enhancing the rigor of this study and informing future research directions. Our specific response is as follows:
The data for this study originates from aquatic product supervision and sampling inspection data publicly released by the market supervision administrations of China's 31 provinces (excluding Hong Kong, Macao, and Taiwan regions) from 2021 to 2023. The core purpose of such data is to serve food safety regulatory disclosures. Consequently, it primarily presents foundational information such as sampling provinces, dates, testing parameters, and compliance criteria. Details regarding the supply chain–including aquaculture origins, transportation routes, and storage conditions–as well as environmental factors near sampling sites (e.g., water pollution levels, farming density) are indeed absent. This gap is a widespread phenomenon within China's current food safety data disclosure system. Addressing this limitation, this study explicitly states in its conclusion section: The absence of critical upstream production information (e.g., aquaculture water quality, feed composition, medication records) and environmental monitoring data prevents a comprehensive analysis of the production–processing–distribution–consumption risk transmission pathway. Based on this, the study proposes future research plans: Integrate data across the entire seafood supply chain from aquaculture to the dining table, extend data collection periods, and incorporate risk analysis for different seafood species; integrate environmental monitoring data (water, sediments, feed, weather, etc.) to establish multi–medium models, and delve into pollutant migration and transformation patterns and their impact on aquatic product risks. This aims to further reveal the supply chain–driven mechanisms behind spatiotemporal distribution patterns through supplementary data dimensions.
All risk assessment results in this study (e.g., risk substance classification, spatio–temporal risk values, spatial autocorrelation characteristics) are strictly tied to specific sampling points. During raw data organization, key sampling point attributes–including sampled provinces, sampling locations, and sampling stages–were preserved. Invalid data lacking provincial labels or with ambiguous sampling locations were excluded. The final valid samples can be precisely mapped to provincial administrative units and specific stage locations, establishing a robust foundation for linking results to sampling points and ensuring traceability of single–point results to specific sampling scenarios. Future efforts will integrate enterprise production records and supply chain traceability platform data to further deepen sampling point coverage, enhancing the accuracy and comprehensiveness of result interpretation.
Comments 4: Another issue, is relying solely on detection frequencies and regulatory non–compliance, which does not fully reflect the actual toxicological risk posed by many contaminants, because their impact on human health depends strongly on chemical form, bioavailability, and metabolic behavior in the body. Different species of metals or organic contaminants exhibit drastically different toxicity profiles; therefore, treating each substance as a single uniform entity may oversimplify the risk assessment. It would be valuable for the authors to consider whether a joint modelling framework, incorporating information on speciation and correlated with this estimated bioavailability, could provide a more biologically meaningful risk classification.
Response 4: We sincerely appreciate your valuable suggestions. We fully concur with your perspective, as you have accurately identified a critical limitation in this study's risk assessment framework: relying solely on pollutant detection frequency and regulatory compliance status for risk determination fails to adequately consider core toxicological characteristics such as chemical form, bioavailability, and metabolic behavior within the body. These factors are precisely what determine the actual health risks. Different forms of metals or pollutants exhibit significant variations in toxicity intensity, exposure pathways, and metabolic transformation patterns. Treating pollutants of diverse forms as a single category in risk assessment simplifies complex toxicological processes, potentially leading to discrepancies between risk assessment outcomes and actual health impacts. As you noted, integrating pollutant form analysis with bioavailability estimation within a combined modeling framework can substantially enhance the biological relevance and scientific rigor of risk classification. We explicitly addressed this limitation in the research constraints section of the paper's conclusions and have identified it as a key focus for future research(line 885–916). In subsequent studies, we will actively incorporate your valuable suggestions: systematically collect morphological analysis data for target pollutants; quantitatively estimate the bioavailability of different pollutant forms using in vitro digestion models or in vivo exposure experiments; and develop a joint modeling framework integrating morphological differences, bioavailability, and pharmacokinetic parameters. This will elevate our assessment from detecting concentration compliance to quantifying actual health risks, making risk classification more aligned with health impacts under real–world exposure scenarios. We extend our most sincere gratitude once again! Your precise and insightful review comments not only helped us clearly identify the study's limitations but also provided invaluable guidance for optimizing future research. This has been crucial in enhancing the academic quality and practical value of this study. We will earnestly incorporate your suggestions, continuously refine our research content, and advance the development of risk assessment methodologies for contaminants in aquatic products.
Comments 5: Furthermore, the dataset does not distinguish between natural and anthropogenic sources of contaminants, which is crucial for interpreting long–term risk patterns and for designing targeted regulatory interventions. The paper would benefit from an expanded discussion of the environmental origins of these substances such as geological background levels, agricultural runoff, industrial discharge, or aquaculture–related inputs and their relative contributions to contamination in aquatic organisms.
Response 5: We sincerely appreciate your valuable suggestions. We have further refined our study in the Discussion section by integrating environmental sources of risk substances and their relative contributions to aquatic contamination. Your feedback has effectively enhanced the rigor and depth of this paper, playing a crucial role in interpreting long–term risk patterns and developing targeted regulatory interventions. All new content is located on lines 670–689 of the manuscript for your reference. All revisions in the manuscript have been marked in red.
Comments 6: Another important limitation is the lack of integration with environmental monitoring data, particularly contaminant levels in water, sediments, and feed. Understanding the correlation between pollutant concentrations in the aquatic environment and their accumulation in organisms is essential to justify the identified spatial patterns and risk trends. Including such comparative data would substantially strengthen the interpretation of spatiotemporal distributions.
Response 6: We sincerely appreciate your valuable suggestions. The issue you raised regarding the lack of integrated environmental monitoring data is critical. We fully agree and understand that the migration and accumulation patterns of environmental pollutants from water, sediments, and feed into aquatic products are essential for explaining and substantiating the spatio–temporal distribution patterns of risks we have identified. We have incorporated additional content in the manuscript's conclusion section to emphasize the importance of integrating environmental monitoring data for this study, specifically located at lines 886–901 of the manuscript. At present, we have not obtained environmental monitoring data. In the next phase, we shall acquire the relevant data to conduct further in–depth research.In future research, we will integrate environmental monitoring data to establish multi–medium models, further investigating pollutant migration and transformation patterns and their impact on aquatic product risks. We sincerely appreciate your suggestions, which are vital for enhancing the depth of this research. All revisions in the manuscript have been marked in red.
Comments 7: Lastly, I recommend adding a dedicated section on the limitations of the chosen methodological approach, explicitly addressing issues such as data completeness, regional reporting discrepancies, detection limit variability, lack of speciation analysis, and the inherent constraints of using secondary surveillance data. Highlighting these aspects would provide greater transparency and help guide future methodological improvements. Please also correct Fig.4.
Response 7: We sincerely appreciate your valuable suggestions, which are crucial for enhancing the transparency of this study and guiding future research optimization. We have added dedicated content to the Conclusions section to supplement the limitations of this study and areas for improvement in further research (specifically located at lines 885–916 of the manuscript), and have comprehensively revised Figure 4. All revisions in the manuscript have been marked in red.
Round 2
Reviewer 2 Report
Comments and Suggestions for Authors
Accepted in the present form.